# Explainable Subgraph Reasoning for Forecasting on Temporal Knowledge Graphs

**Zhen Han**[*1,2], **Peng Chen**[*2,3], **Yunpu Ma**[†1], **Volker Tresp**[†1,2]
[1]Institute of Informatics, LMU Munich  [2] Corporate Technology, Siemens AG
[3]Department of Informatics, Technical University of Munich
`zhen.han@campus.lmu.de, peng.chen@tum.de`
`cognitive.yunpu@gmail.com, volker.tresp@siemens.com`

## Abstract

Modeling time-evolving knowledge graphs (KGs) has recently gained increasing interest. Here, graph representation learning has become the dominant paradigm for link prediction on temporal KGs. However, the embedding-based approaches largely operate in a black-box fashion, lacking the ability to interpret their predictions. This paper provides a link forecasting framework that reasons over query-relevant subgraphs of temporal KGs and jointly models the structural dependencies and the temporal dynamics. Especially, we propose a temporal relational attention mechanism and a novel reverse representation update scheme to guide the extraction of an enclosing subgraph around the query. The subgraph is expanded by an iterative sampling of temporal neighbors and by attention propagation. Our approach provides human-understandable evidence explaining the forecast. We evaluate our model on four benchmark temporal knowledge graphs for the link forecasting task. While being more explainable, our model obtains a relative improvement of up to 20 % on Hits@1 compared to the previous best temporal KG forecasting method. We also conduct a survey with 53 respondents, and the results show that the evidence extracted by the model for link forecasting is aligned with human understanding.

## 1 Introduction

Reasoning, a process of inferring new knowledge from available facts, has long been considered an essential topic in AI research. Recently, reasoning on knowledge graphs (KG) has gained increasing interest (Das et al., 2017; Ren et al., 2020; Hildebrandt et al., 2020). A knowledge graph is a graph-structured knowledge base that stores factual information in the form of triples $(s, p, o)$, e.g., (*Alice, livesIn, Toronto*). In particular, $s$ (subject) and $o$ (object) are expressed as nodes and $p$ (predicate) as an edge type. Most knowledge graph models assume that the underlying graph is static. However, in the real world, facts and knowledge can change with time. For example, (*Alice, livesIn, Toronto*) becomes invalid after Alice moves to Vancouver. To accommodate time-evolving multi-relational data, temporal KGs have been introduced (Boschee et al., 2015), where a temporal fact is represented as a quadruple by extending the static triple with a timestamp $t$ indicating the triple is valid at $t$, i.e. (*Barack Obama, visit, India*, 2010-11-06).

In this work, we focus on forecasting on temporal KGs, where we infer future events based on past events. Forecasting on temporal KGs can improve a plethora of downstream applications such as decision support in personalized health care and finance. The use cases often require the predictions made by the learning models to be interpretable, such that users can understand and trust the predictions. However, current machine learning approaches (Trivedi et al., 2017; Jin et al., 2019) for temporal KG forecasting operate in a black-box fashion, where they design an embedding-based score function to estimate the plausibility of a quadruple. These models cannot clearly show which evidence contributes to a prediction and lack explainability to the forecast, making them less suitable for many real-world applications.

---

[*]Equal contribution.
[†]Corresponding authors.

Explainable approaches can generally be categorized into post-hoc interpretable methods and integrated transparent methods (Došilović et al., 2018). Post-hoc interpretable approaches (Montavon et al., 2017; Ying et al., 2019) aim to interpret the results of a black-box model, while integrated transparent approaches (Das et al., 2017; Qiu et al., 2019; Wang et al., 2019) have an explainable internal mechanism. In particular, most integrated transparent (Lin et al., 2018; Hildebrandt et al., 2020) approaches for KGs employ path-based methods to derive an explicit reasoning path and demonstrate a transparent reasoning process. The path-based methods focus on finding the answer to a query within a single reasoning chain. However, many complicated queries require multiple supporting reasoning chains rather than just one reasoning path. Recent work (Xu et al., 2019; Teru et al., 2019) has shown that reasoning over local subgraphs substantially boosts performance while maintaining interpretability. However, these explainable models cannot be applied to temporal graph-structured data because they do not take time information into account. This work aims to design a transparent forecasting mechanism on temporal KGs that can generate informative explanations of the predictions.

In this paper, we propose an **explainable reasoning** framework for forecasting future links on **t**emporal knowl**e**dge graphs, xERTE, which employs a sequential reasoning process over local subgraphs. To answer a query in the form of (subject $e_q$, predicate $p_q$, ?, timestamp $t_q$), xERTE starts from the query subject, iteratively samples relevant edges of entities included in the subgraph and propagates attention along the sampled edges. After several rounds of expansion and pruning, the missing object is predicted from entities in the subgraph. Thus, the extracted subgraph can be seen as a concise and compact graphical explanation of the prediction. To guide the subgraph to expand in the direction of the query's interest, we propose a temporal relational graph attention (TRGA) mechanism. We pose temporal constraints on passing messages to preserve the causal nature of the temporal data. Specifically, we update the time-dependent hidden representation of an entity $e_i$ at a timestamp $t$ by attentively aggregating messages from its temporal neighbors that were linked with $e_i$ prior to $t$. We call such temporal neighbors as prior neighbors of $e_i$. Additionally, we use an embedding module consisting of stationary entity embeddings and functional time encoding, enabling the model to capture both global structural information and temporal dynamics. Besides, we develop a novel representation update mechanism to mimic human reasoning behavior. When humans perform a reasoning process, their perceived profiles of observed entities will update, as new clues are found. Thus, it is necessary to ensure that all entities in a subgraph can receive messages from prior neighbors newly added to the subgraph. To this end, the proposed representation update mechanism enables every entity to receive messages from its farthest prior neighbors in the subgraph.

The major contributions of this work are as follows. **(1)** We develop xERTE, the first explainable model for predicting *future links* on temporal KGs. The model is based on a temporal relational attention mechanisms that preserves the causal nature of the temporal multi-relational data. **(2)** Unlike most black-box embedding-based models, xERTE visualizes the reasoning process and provides an interpretable inference graph to emphasize important evidence. **(3)** The dynamical pruning procedure enables our model to perform reasoning on large-scale temporal knowledge graphs with millions of edges. **(4)** We apply our model for forecasting future links on four benchmark temporal knowledge graphs. The results show that our method achieves on average a better performance than current state-of-the-art methods, thus providing a new baseline. **(5)** We conduct a survey with 53 respondents to evaluate whether the extracted evidence is aligned with human understanding.

## 2    RELATED WORK

Representation learning is an expressive and popular paradigm underlying many KG models. The embedding-based approaches for knowledge graphs can generally be categorized into bilinear models (Nickel et al., 2011; Yang et al., 2014; Ma et al., 2018a), translational models (Bordes et al., 2013; Lv et al., 2018; Sun et al., 2019; Hao et al., 2019), and deep-learning models (Dettmers et al., 2017; Schlichtkrull et al., 2018). However, the above methods are not able to use rich dynamics available on temporal knowledge graphs. To this end, several studies have been conducted for temporal knowledge graph reasoning (García-Durán et al., 2018; Ma et al., 2018b; Jin et al., 2019; Goel et al., 2019; Lacroix et al., 2020; Han et al., 2020a;b; Zhu et al., 2020). The published approaches are largely black-box, lacking the ability to interpret their predictions. Recently, several explainable reasoning methods for knowledge graphs have been proposed (Das et al., 2017; Xu et al., 2019;

Hildebrandt et al., 2020; Teru et al., 2019). However, the above explainable methods can only deal with static KGs, while our model is designed for interpretable forecasting on temporal KGs.

## 3 PRELIMINARIES

Let $\mathcal{E}$ and $\mathcal{P}$ represent a finite set of entities and predicates, respectively. A temporal knowledge graph is a collection of timestamped facts written as quadruples. A quadruple $q = (e_s, p, e_o, t)$ represents a timestamped and labeled edge between a subject entity $e_s \in \mathcal{E}$ and an object entity $e_o \in \mathcal{E}$, where $p \in \mathcal{P}$ denotes the edge type (predicate). The temporal knowledge graph forecasting task aims to predict unknown links at future timestamps based on observed past events.

**Definition 1** *(Temporal KG forecasting). Let $\mathcal{F}$ represent the set of all ground-truth quadruples, and let $(e_q, p_q, e_o, t_q) \in \mathcal{F}$ denote the target quadruple. Given a query $(e_q, p_q, ?, t_q)$ derived from the target quadruple and a set of observed prior facts $\mathcal{O} = \{(e_i, p_k, e_j, t_l) \in \mathcal{F} | t_l < t_q\}$, the temporal KG forecasting task is to predict the missing object entity $e_o$. Specifically, we consider all entities in the set $\mathcal{E}$ as candidates and rank them by their likelihood to form a true quadruple together with the given subject-predicate-pair at timestamp $t_q$*[1].

For a given query $q = (e_q, p_q, ?, t_q)$, we build an *inference graph* $\mathcal{G}_{inf}$ to visualize the reasoning process. Unlike in temporal KGs, where a node represents an entity, each node in $\mathcal{G}_{inf}$ is an entity-timestamp pair. The inference graph is a directed graph in which a link points from a node with an earlier timestamp to a node with a later timestamp.

**Definition 2** *(Node in Inference Graph and its Temporal Neighborhood). Let $\mathcal{E}$ represent all entities, $\mathcal{F}$ denote all ground-truth quadruples, and let $t$ represent a timestamp. A node in an inference graph $\mathcal{G}_{inf}$ is defined as an entity-timestamp pair $v = (e_i, t), e_i \in \mathcal{E}$. We define the set of one-hop prior neighbors of $v$ as $\mathcal{N}_{v=(e_i,t)} = \{(e_j, t') | (e_i, p_k, e_j, t') \in \mathcal{F} \wedge (t' < t)\}$*[2]*. For simplicity, we denote one-hop prior neighbors as $\mathcal{N}_v$. Similarly, we define the set of one-hop posterior neighbors of $v$ as $\overline{\mathcal{N}}_{v=(e_i,t)} = \{(e_j, t') | (e_j, p_k, e_i, t) \in \mathcal{F} \wedge (t' > t)\}$. We denote them as $\overline{\mathcal{N}}_v$ for short.*

We provide an example in Figure 4 in the appendix to illustrate the inference graph.

## 4 OUR MODEL

We describe xERTE in a top-down fashion where we provide an overview in Section 4.1 and then explain each module from Section 4.2 to 4.6.

### 4.1 SUBGRAPH REASONING PROCESS

Our model conducts the reasoning process on a dynamically expanded inference graph $\mathcal{G}_{inf}$ extracted from the temporal KG. We show a toy example in Figure 1. Given query $q = (e_q, p_q, ?, t_q)$, we initialize $\mathcal{G}_{inf}$ with node $v_q = (e_q, t_q)$ consisting of the query subject and the query time. The inference graph expands by sampling prior neighbors of $v_q$. For example, suppose that $(e_q, p_k, e_j, t')$ is a valid quadruple where $t' < t_q$, we add the node $v_1 = (e_j, t')$ into $\mathcal{G}_{inf}$ and link it with $v_q$ where the link is labeled with $p_k$ and points from $v_q$ to $v_1$. We use an embedding module to assign each node and predicate included in $\mathcal{G}_{inf}$ a temporal embedding that is shared across queries. The main goal of the embedding module is to let the nodes access query-independent information and get a broad view of the graph structure since the following temporal relational graph attention (TRGA) layer only performs query-dependent message passing locally. Next, we feed the inference graph into the TRGA layer that takes node embeddings and predicate embeddings as the input, produces a query-dependent representation for each node by passing messages on the small inference graph, and computes a query-dependent attention score for each edge. As explained in Section 4.7, we propagate the attention of each node to its prior neighbors using the edge attention scores. Then we further expand $\mathcal{G}_{inf}$ by sampling the prior neighbors of the nodes in $\mathcal{G}_{inf}$. The expansion will grow

---

[1]Throughout this work, we add reciprocal relations for every quadruple, i.e., we add $(e_o, p^{-1}, e_s, t)$ for every $(e_s, p, e_o, t)$. Hence, the restriction to predict object entities does not lead to a loss of generality.

[2]Prior neighbors linked with $e_i$ as subject entity, e.g., $(e_j, p_k, e_i, t)$, are covered using reciprocal relations.

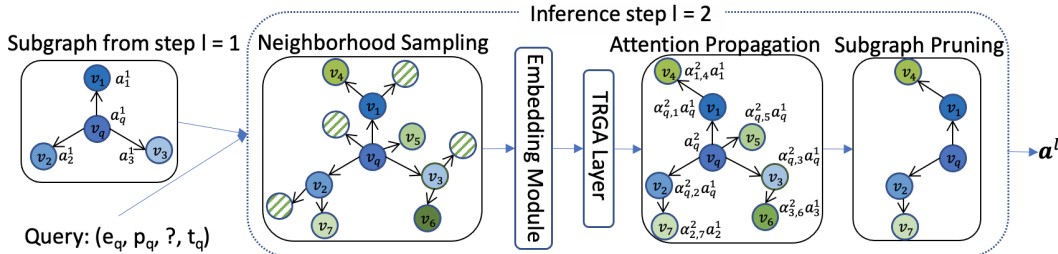

Figure 1: Model Architecture. We take the second inference step ($l = 2$) as an example. Each directed edge points from a source node to its prior neighbor. ⊘ denotes nodes that have not been sampled. $a_i^l$ means the attention score of node $v_i$ at the $l^{th}$ inference step. $\alpha_{i,j}^l$ is the attention score of the edge between node $i$ and its prior neighbor $j$ at the $l^{th}$ inference step. Note that all scores are query-dependent. For simplicity, we do not show edge labels (predicates) in the figure.

rapidly and cover almost all nodes after a few steps. To prevent the inference graph from exploding, we reduce the edge amount by pruning the edges that gain less attention. As the expansion and pruning iterate, $\mathcal{G}_{inf}$ allocates more and more information from the temporal KG. After running $L$ inference steps, the model selects the entity with the highest attention score in $\mathcal{G}_{inf}$ as the prediction of the missing query object, where the inference graph itself serves as a graphical explanation.

## 4.2 NEIGHBORHOOD SAMPLING

We define the set of edges between node $v = (e_i, t)$ and its prior neighbors $\mathcal{N}_v$ as $\mathcal{Q}_v$, where $q_v \in \mathcal{Q}_v$ is a *prior edge* of $v$. To reduce the complexity, we sample a subset of prior edges $\hat{\mathcal{Q}}_v \in \mathcal{Q}_v$ at each inference step. We denote the remaining prior neighbors and posterior neighbors of node $v$ after the sampling as $\hat{\mathcal{N}}_v$ and $\overline{\hat{\mathcal{N}}_v}$, respectively. Note that there might be multiple edges between node $v$ and its prior neighbor $u$ because of multiple predicates. If there is at least one edge that has been sampled between $v$ and $u$, we add $u$ into $\hat{\mathcal{N}}_v$. The sampling can be uniform if there is no bias, it can also be temporally biased using a non-uniform distribution. For instance, we may want to sample more edges closer to the current time point as the events that took place long ago may have less impact on the inference. Specifically, we propose three different sampling strategies: (1) **Uniform sampling**. Each prior edge $q_v \in \mathcal{Q}_v$ has the same probability of being selected: $\mathbb{P}(q_v) = 1/|\mathcal{Q}_v|$. (2) **Time-aware exponentially weighted sampling**. We temporally bias the neighborhood sampling using an exponential distribution and assign the probability $\mathbb{P}(q_v = (e_i, p_k, e_j, t')) = \exp(t' - t)/\sum_{(e_i, p_l, e_m, t'') \in \mathcal{Q}_v} \exp(t'' - t)$ to each prior neighbor, which negatively correlates with the time difference between node $v$ and its prior neighbor $(e_j, t')$. Note that $t'$ and $t''$ are prior to $t$. (3) **Time-aware linearly weighted sampling**. We use a linear function to bias the sampling. Compared to the second strategy, the quadruples occurred in early stages have a higher probability of being sampled. Overall, we have empirically found that the second strategy is most beneficial to our framework and provide a detailed ablation study in Section 5.2.

## 4.3 EMBEDDING

In temporal knowledge graphs, graph structures are no longer static, as entities and their links evolve over time. Thus, entity features may change and exhibit temporal patterns. In this work, the embedding of an entity $e_i \in \mathcal{E}$ at time $t$ consists of a static low-dimensional vector and a functional representation of time. The time-aware entity embedding is defined as $\mathbf{e}_i(t) = [\bar{\mathbf{e}}_i || \mathbf{\Phi}(t)]^T \in \mathbb{R}^{d_S + d_T}$. Here, $\bar{\mathbf{e}}_i \in R^{d_S}$ represents the static embedding that captures time-invariant features and global dependencies over the temporal KG. $\mathbf{\Phi}(\cdot)$ denotes a time encoding that captures temporal dependencies between entities (Xu et al., 2020). We provide more details about $\mathbf{\Phi}(\cdot)$ in Appendix I. $||$ denotes the concatenation operator. $d_S$ and $d_T$ represent the dimensionality of the static embedding and the time embedding, which can be tuned according to the temporal fraction of the given dataset. We also tried the temporal encoding presented in Goel et al. (2019), which has significantly more parameters. But we did not see considerable improvements. Besides, we assume that predicate features do not evolve. Thus, we learn a stationary embedding vector $\mathbf{p}_k$ for each predicate $p_k$.

## 4.4 TEMPORAL RELATIONAL GRAPH ATTENTION LAYER

Here, we propose a temporal relational graph attention (TRGA) layer for identifying the relevant evidence in the inference graph related to a given query $q$. The input to the TRGA layer is a set of entity embeddings $\mathbf{e}_i(t)$ and predicate embeddings $\mathbf{p}_k$ in the given inference graph. The layer produces a query-dependent attention score for each edge and a new set of hidden representations as its output. Similar to GraphSAGE (Hamilton et al., 2017) and GAT (Veličković et al., 2017), the TRGA layer performs a local representation aggregation. To avoid misusing future information, we only allow message passing from prior neighbors to posterior neighbors. Specifically, for each node $v$ in the inference graph, the aggregation function fuses the representation of node $v$ and the sampled prior neighbors $\hat{\mathcal{N}}_v$ to output a time-aware representation for $v$. Since entities may play different roles, depending on the predicate they are associated with, we incorporate the predicate embeddings in the attention function to exploit relation information. Instead of treating all prior neighbors with equal importance, we take the query information into account and assign varying importance levels to each prior neighbor $u \in \hat{\mathcal{N}}_v$ by calculating a query-dependent attention score using

$$e^l_{vu}(q, p_k) = \mathbf{W}^l_{sub}(\mathbf{h}^{l-1}_v||\mathbf{p}^{l-1}_k||\mathbf{h}^{l-1}_{e_q}||\mathbf{p}^{l-1}_q)\mathbf{W}^l_{obj}(\mathbf{h}^{l-1}_u||\mathbf{p}^{l-1}_k||\mathbf{h}^{l-1}_{e_q}||\mathbf{p}^{l-1}_q), \tag{1}$$

where $e^l_{vu}(q, p_k)$ is the attention score of the edge $(v, p_k, u)$ regarding the query $q = (e_q, p_q, ?, t_q)$, $p_k$ corresponds to the predicate between node $u$ and node $v$, $\mathbf{p}_k$ and $\mathbf{p}_q$ are predicate embeddings. $\mathbf{h}^{l-1}_v$ denotes the hidden representation of node $v$ at the $(l-1)^{th}$ inference step. When $l = 1$, i.e., for the first layer, $\mathbf{h}^0_v = \mathbf{W}_v\mathbf{e}_i(t) + \mathbf{b}_v$, where $v = (e_i, t)$. $\mathbf{W}^l_{sub}$ and $\mathbf{W}^l_{obj}$ are two weight matrices for capturing the dependencies between query features and node features. Then, we compute the normalized attention score $\alpha^l_{vu}(q, p_k)$ using the *softmax function* as follows

$$\alpha^l_{vu}(q, p_k) = \frac{\exp(e^l_{vu}(q, p_k))}{\sum_{w \in \hat{\mathcal{N}}_v} \sum_{p_z \in \mathcal{P}_{vw}} \exp(e^l_{vw}(q, p_z))}, \tag{2}$$

where $\mathcal{P}_{vw}$ represents the set of labels of edges that connect nodes $v$ and $w$. Once obtained, we aggregate the representations of prior neighbors and weight them using the normalized attention scores, which is written as

$$\widetilde{\mathbf{h}}^l_v(q) = \sum_{u \in \hat{\mathcal{N}}_v} \sum_{p_k \in \mathcal{P}_{vu}} \alpha^l_{vu}(q, p_k)\mathbf{h}^{l-1}_u(q). \tag{3}$$

We combine the hidden representation $\mathbf{h}^{l-1}_v(q)$ of node $v$ with the aggregated neighborhood representation $\widetilde{\mathbf{h}}^l_v(q)$ and feed them into a fully connected layer with a LeakyReLU activation function $\sigma(\cdot)$, as shown below

$$\mathbf{h}^l_v(q) = \sigma(\mathbf{W}^l_h(\gamma\mathbf{h}^{l-1}_v(q) + (1-\gamma)\widetilde{\mathbf{h}}^l_v(q) + \mathbf{b}^l_h)), \tag{4}$$

where $\mathbf{h}^l_v(q)$ denotes the representation of node $v$ at the $l^{th}$ inference step, and $\gamma$ is a hyperparameter. Further, we use the same layer to update the relation embeddings, which is of the form $\mathbf{p}^l_k = \mathbf{W}^l_h\mathbf{p}^{l-1}_k + \mathbf{b}^l_h$. Thus, the relations are projected to the same embedding space as nodes and can be utilized in the next inference step.

## 4.5 ATTENTION PROPAGATION AND SUBGRAPH PRUNING

After having the edges' attention scores in the inference graph, we compute the attention score $a^l_{v,q}$ of node $v$ regarding query $q$ at the $l^{th}$ inference step as follows:

$$a^l_{v,q} = \sum_{u \in \widetilde{\mathcal{N}}_v} \sum_{p_z \in \mathcal{P}_{uv}} \alpha^l_{uv}(q, p_z)a^{l-1}_{u,q}. \tag{5}$$

Thus, we propagate the attention of each node to its prior neighbors. As stated in Definition 2, each node in inference graph is an entity-timestamp pair. To assign each entity a unique attention score, we aggregate the attention scores of nodes whose entity is the same:

$$a^l_{e_i,q} = g(a^l_{v,q}|v(e) = e_i), \quad \text{for } v \in \mathcal{V}_{\mathcal{G}_{inf}}, \tag{6}$$

where $a_{e_i,q}^l$ denotes the attention score of entity $e_i$, $\mathcal{V}_{\mathcal{G}_{inf}}$ is the set of nodes in inference graph $\mathcal{G}_{inf}$. $v(e)$ represents the entity included in node $v$, and $g(\cdot)$ represents a score aggregation function. We try two score aggregation functions $g(\cdot)$, i.e., summation and mean. We conduct an ablation study and find that the summation aggregation performs better. To demonstrate which evidence is important for the reasoning process, we assign each edge in the inference graph a contribution score. Specifically, the contribution score of an edge $(v, p_k, u)$ is defined as $c_{vu}(q, p_k) = \alpha_{vu}^l(q, p_k) a_{v,q}^l$, where node $u$ is a prior neighbor of node $v$ associated with the predicate $p_k$. We prune the inference graph at each inference step and keep the edges with $K$ largest contribution scores. We set the attention score of entities, which the inference graph does not include, to zero. Finally, we rank all entity candidates according to their attention scores and choose the entity with the highest score as our prediction.

### 4.6 REVERSE REPRESENTATION UPDATE MECHANISM

When humans perform a reasoning process, the perceived profile of an entity during the inference may change as new evidence joins the reasoning process. For example, we want to predict the profitability of company A. We knew that A has the largest market portion, which gives us a high expectation about A's profitability. However, a new evidence shows that conglomerate B enters this market as a strong competitor. Although the new evidence is not directly related to A, it indicates that there will be a high competition between A and B, which lowers our expectation about A's profitability. To mimic human reasoning behavior, we should ensure that all existing nodes in inference graph $\mathcal{G}_{inf}$ can receive messages from nodes newly added to $\mathcal{G}_{inf}$. However, since $\mathcal{G}_{inf}$ expands once at each inference step, it might include $l$-hop neighbors of the query subject at the $l^{th}$ step. The vanilla solution is to iterate the message passing $l$ times at the $l^{th}$ inference step, which means that we need to run the message passing $(1 + L) \cdot L/2$ times in total, for $L$ inference steps. To avoid the quadratic increase of message passing iterations, we propose a novel reverse representation update mechanism. Recall that, to avoid violating temporal constraints, we use prior neighbors to update nodes' representations. And at each inference step, we expand $\mathcal{G}_{inf}$ by adding prior neighbors of each node in $\mathcal{G}_{inf}$. For example, assuming that we are at the fourth inference step, for a node that has been added at the second step, we only need to aggregate messages from nodes added at the third and fourth steps. Hence, we can update the representations of nodes in reverse order as they have been added in $\mathcal{G}_{inf}$. Specifically, at the $l^{th}$ inference step, we first update the representations of nodes added at the $(l-1)^{th}$ inference step, then the nodes added at $(l-2)^{th}$, and so forth until $l = 0$, as shown in Algorithm 1 in the appendix. In this way, we compute messages along each edge in $\mathcal{G}_{inf}$ only once and ensure that every node can receive messages from its farthest prior neighbor.

### 4.7 LEARNING

We split quadruples of a temporal KG into *train*, *validation*, and *test* sets by timestamps, ensuring (timestamps of training set)<(timestamps of validation set)<(timestamps of test set). We use the binary cross-entropy as the loss function, which is defined as

$$\mathcal{L} = -\frac{1}{|\mathcal{Q}|} \sum_{q \in \mathcal{Q}} \frac{1}{|\mathcal{E}_q^{inf}|} \sum_{e_i \in \mathcal{E}_q^{inf}} \left( y_{e_i,q} \log(\frac{a_{e_i,q}^L}{\sum_{e_j \in \mathcal{E}_q^{inf}} a_{e_j,q}^L}) + (1 - y_{e_i,q}) \log(1 - \frac{a_{e_i,q}^L}{\sum_{e_j \in \mathcal{E}_q^{inf}} a_{e_j,q}^L}) \right),$$

where $\mathcal{E}_q^{inf}$ represents the set of entities in the inference graph of the query $q$, $y_{e_i,q}$ represents the binary label that indicates whether $e_i$ is the answer for $q$, and $Q$ denotes the set of training quadruples. $a_{e_i,q}^L$ denotes the attention score of $e_i$ at the final inference step. We list all model parameters in Table 2 in the appendix. Particularly, we jointly learn the embeddings and other model parameters by end-to-end training.

## 5 EXPERIMENTS

### 5.1 DATASETS AND BASELINES

Integrated Crisis Early Warning System (ICEWS) (Boschee et al., 2015) and YAGO (Mahdisoltani et al., 2013) have established themselves in the research community as benchmark datasets of temporal KGs. The ICEWS dataset contains information about political events with time annotations,

e.g., (*Barack Obama, visit, Malaysia,* 2014-02-19). We evaluate our model on three subsets of the ICEWS dataset, i.e., ICEWS14, ICEWS18, and ICEWS05-15, that contain event facts in 2014, 2018, and the facts from 2005 to 2015, respectively. The YAGO dataset is a temporal knowledge base that fuses information from Wikipedia with the English WordNet dataset (Miller, 1995). Following the experimental settings of HyTE (Dasgupta et al., 2018), we use a subset and only deal with year level granularity by dropping the month and date information. We compare our approach and baseline methods by performing the link prediction task on the ICEWS14, ICEWS18, ICEWS0515, and YAGO datasets. The statistics of the datasets are provided in Appendix C.

We compare xERTE with benchmark temporal KG and static KG reasoning models. From the temporal KG reasoning models, we compare our model with several state-of-the-art methods, including TTransE (Leblay & Chekol, 2018), TA-DistMult/TA-TransE (García-Durán et al., 2018), DE-SimplE(Goel et al., 2019), TNTComplEx (Lacroix et al., 2020), CyGNet(Zhu et al., 2020), and RE-Net (Jin et al., 2019). From the static KG reasoning models, we choose TransE (Bordes et al., 2013), DistMult (Yang et al., 2014), and ComplEx (Trouillon et al., 2016).

## 5.2 EXPERIMENTAL RESULTS AND ABLATION STUDY

| Datasets | ICEWS14 - filtered | | | | ICEWS05-15 - filtered | | | | ICEWS18 - filtered | | | | YAGO - filtered | | | |
|---|---|---|---|---|---|---|---|---|---|---|---|---|---|---|---|---|
| Model | MRR | HITS@1 | HITS@3 | HITS@10 | MRR | Hits@1 | Hits@3 | Hits@10 | MRR | Hits@1 | Hits@3 | Hits@10 | MRR | Hits@1 | Hits@3 | Hits@10 |
| TransE | 22.48 | 13.36 | 25.63 | 41.23 | 22.55 | 13.05 | 25.61 | 42.05 | 12.24 | 5.84 | 12.81 | 25.10 | 11.69 | 10.37 | 11.96 | 13.83 |
| DistMult | 27.67 | 18.16 | 31.15 | 46.96 | 28.73 | 19.33 | 32.19 | 47.54 | 10.17 | 4.52 | 10.33 | 21.25 | 11.98 | 10.20 | 12.31 | 14.93 |
| ComplEx | 30.84 | 21.51 | 34.48 | 49.58 | 31.69 | 21.44 | 35.74 | 52.04 | 21.01 | 11.87 | 23.47 | 39.87 | 12.07 | 10.42 | 12.36 | 14.82 |
| TTransE | 13.43 | 3.11 | 17.32 | 34.55 | 15.71 | 5.00 | 19.72 | 38.02 | 8.31 | 1.92 | 8.56 | 21.89 | 5.68 | 1.42 | 9.04 | 11.21 |
| TA-DistMult | 26.47 | 17.09 | 30.22 | 45.41 | 24.31 | 14.58 | 27.92 | 44.21 | 16.75 | 8.61 | 18.41 | 33.59 | 11.50 | 10.21 | 11.90 | 13.88 |
| TA-TransE | 17.41 | 0.00 | 29.19 | 47.41 | 19.37 | 1.81 | 31.34 | 50.33 | 12.59 | 0.01 | 17.92 | 37.38 | 6.74 | 2.13 | 11.01 | 12.28 |
| DE-SimplE | 32.67 | 24.43 | 35.69 | 49.11 | 35.02 | 25.91 | 38.99 | 52.75 | 19.30 | 11.53 | 21.86 | 34.80 | 11.73 | 10.70 | 12.10 | 13.51 |
| TNTComplEx | 32.12 | 23.35 | 36.03 | 49.13 | 27.54 | 19.52 | 30.80 | 42.86 | 21.23 | 13.28 | 24.02 | 36.91 | 12.00 | 11.12 | 12.13 | 13.57 |
| CyGNet[3] | 32.73 | 23.69 | 36.31 | 50.67 | 34.97 | 25.67 | 39.09 | 52.94 | 24.93 | 15.90 | 28.28 | 42.61 | 12.48 | 11.00 | 12.66 | 14.82 |
| RE-Net | 38.28 | 28.68 | 41.34 | 54.52 | 42.97 | 31.26 | 46.85 | 63.47 | 28.81 | 19.05 | 32.44 | **47.51** | 54.87 | 47.51 | 57.84 | **65.81** |
| xERTE | **40.79** | **32.70** | **45.67** | **57.30** | **46.62** | **37.84** | **52.31** | **63.92** | **29.31** | **21.03** | **33.51** | 46.48 | 53.62 | **48.53** | **58.42** | 60.53 |

Table 1: Results of future link prediction on four datasets. Compared metrics are **time-aware** filtered MRR (%) and Hits@1/3/10 (%). The best results among all models are in bold.

**Comparison results** Table 1 summarizes the **time-aware** filtered results of the link prediction task on the ICEWS and YAGO datasets[4]. The time-aware filtering scheme only filters out triples that are genuine at the query time while the filtering scheme applied in prior work (Jin et al., 2019; Zhu et al., 2020) filters all triples that occurred in history. A detailed explanation is provided in Appendix D. Overall, xERTE outperforms all baseline models on ICEWS14/05-15/18 in MRR and Hits@1/3/10 while being more interpretable. Compared to the strongest baseline RE-Net, xERTE obtains a relative improvement of 5.60% and 15.15% in MRR and Hits@1, which are averaged on ICEWS14/05-15/18. Especially, xERTE achieves more gains in Hits@1 than in Hits@10. It confirms the assumption that subgraph reasoning helps xERTE make a sharp prediction by exploiting local structures. On the YAGO dataset, xERTE achieves comparable results with RE-Net in terms of MRR and Hits@1/3. To assess the importance of each component, we conduct several ablation studies and show their results in the following.

**Representation update analysis** We train a model without the reverse representation update mechanism to investigate how this mechanism contributes to our model. Since the reverse representation update ensures that each node can receive messages from all its prior neighbors in the inference graph, we expect this mechanism could help nodes mine available information. This update mechanism should be especially important for nodes that only have been involved in a small number of events. Since the historical information of such nodes is quite limited, it is very challenging to forecast their future behavior. In Figure 2a and 2b we show the metrics of Hits@1 and Hits@10 against the number of nodes in the inference graph. It can be observed the model with the reverse update mechanism performs better in general. In particular, this update mechanism significantly improves the performance if the query subject only has a small number of neighbors in the subgraph, which meets our expectation.

---

[3]We found that CyGNet does not perform subject prediction in its evaluation code and does not report time-aware filtered results. The performance significantly drops after fixing the code.

[4]Code and datasets are available at https://github.com/TemporalKGTeam/xERTE

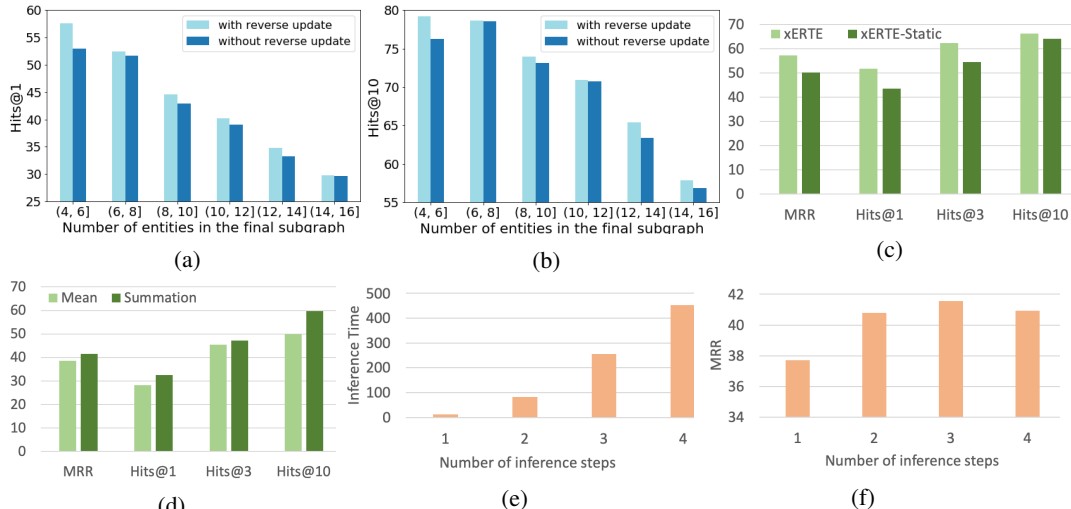

Figure 2: Ablation Study. Unlike in Table 1 that reports results on the whole test set, here we filter out test quadruples that contain unseen entities. (a)-(b) We compare the model with/without the reverse representation update in terms of raw Hits@1(%) and Hits@10(%) on ICEWS14, respectively. (c) Temporal embedding analysis on YAGO. We refer the model without temporal embeddings as xERTE-Static. (d) Attention score aggregation function analysis on ICEWS14: raw MRR (%) and Hits@1/3/10(%). (e) Inference time (seconds) on the test set of ICEWS14 regarding different inference step settings $L \in \{1, 2, 3, 4\}$. (f) Raw MRR(%) on ICEWS14 regarding different inference step settings $L$.

**Time-aware representation analysis and node attention aggregation**   To verify the importance of the time embedding, we evaluate the performance of a model without time encoding. As shown in Figure 2c, removing the time-dependent part from entity representations sacrifices the model's performance significantly. Recall that each node in inference graph $\mathcal{G}_{inf}$ is associated with a timestamp, the same entity might appear in several nodes in $\mathcal{G}_{inf}$ with different timestamps. To get a unified attention score for each entity, we aggregate the attention scores of nodes whose entity is the same. Figure 2d shows that the summation aggregator brings a considerable gain on ICEWS14.

**Sampling analysis**   We run experiments with different sampling strategies proposed in Section 4.2. To assess the necessity of the time-aware weighted sampling, we propose a deterministic version of the time-aware weighted sampling, where we chronologically sort the prior edges of node $v$ in terms of their timestamps and select the last $N$ edges to build the subset $\hat{\mathcal{Q}}_v$. The experimental results are provided in Table 3 in the appendix. We find that the sampling strategy has a considerable influence on model's performance. Sampling strategies that bias towards recent quadruples perform better. Specifically, the exponentially time-weighted strategy performs better than the linear time-weighted strategy and the deterministic last-N-edges strategy.

**Time cost analysis**   The time cost of xERTE is affected not only by the scale of a dataset but also by the number of inference steps $L$. Thus, we run experiments of inference time and predictive power regarding different settings of $L$ and show the results in Figures 2e and 2f. We see that the model achieves the best performance with $L = 3$ while the training time significantly increases as $L$ goes up. To make the computation more efficient, we develop a series of segment operations for subgraph reasoning. Please see Appendix G for more details.

### 5.3   GRAPHICAL EXPLANATION AND HUMAN EVALUATION

The extracted inference graph provides a graphical explanation for model's prediction. As introduced in 4.7, we assign each edge in the inference graph a contribution score. Thus, users can trace back the important evidence that the prediction mainly depends on. We study a query chosen from the test set, where we predict whom will Catherine Ashton visit on Nov. 9, 2014 and show the final

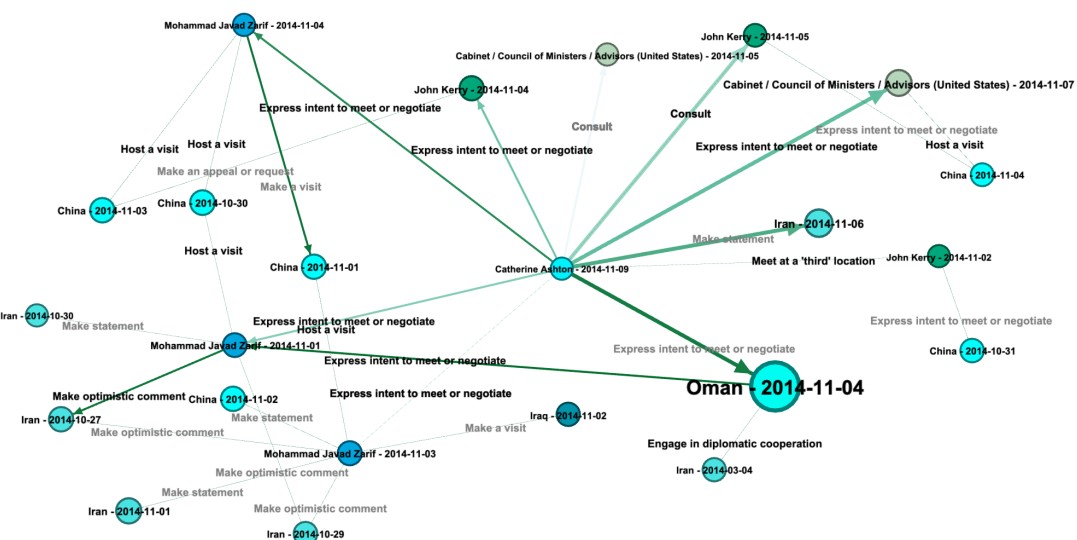

Figure 3: The inference graph for the query (*Catherine Ashton, Make a visit, ?,* 2014-11-09) from ICEWS14. The biggest cyan node represents the object predicted by xERTE. The cyan node with the entity *Catherine Ashton* and the timestamp 2014-11-09 represents the given query subject and the query timestamp. The node size indicates the value of the node attention score. Also, the edges' color indicates the contribution score of the edge, where darkness increases as the contribution score goes up. The entity at an arrow's tail, the predicate on the arrow, the entity and the timestamp at the arrow's head build a true quadruple.

inference graph in Figure 3. In this case, the model's prediction is Oman. And (*Catherine Ashton, express intent to meet or negotiate, Oman,* 2014-11-04) is the most important evidence to support this answer.

To assess whether the evidence is informative for users in an objective setting, we conduct a survey where respondents evaluate the relevance of the extracted evidence to the prediction. More concretely, we set up an online quiz consisting of 7 rounds. Each round is centered around a query sampled from the test set of ICEWS14/ICEWS05-15. Along with the query and the ground-truth answer, we present the human respondents with two pieces of evidence in the inference graph with high contribution scores and two pieces of evidence with low contribution scores in a randomized order. Specifically, each evidence is based on a chronological reasoning path that connects the query subject with an object candidate. For example, given a query (*police, arrest, ?,* 2014-12-28), an extracted clue is that police made statements to lawyers on 2014-12-08, then lawyers were criticized by citizens on 2014-12-10. In each round, we set up three questions to ask the participants to choose the most relevant evidence, the most irrelevant evidence, and sort the pieces of evidence according to their relevance. Then we rank the evidence according to the contribution scores computed by our model and check whether the relevance order classified by the respondents matches that estimated by our models. We surveyed 53 participants, and the average accuracy of all questions is 70.5%. Moreover, based on a majority vote, 18 out of 21 questions were answered correctly, indicating that the extracted inference graphs are informative, and the model is aligned with human intuition. The complete survey and a detailed evaluation are reported in Appendix H.

## 6  CONCLUSION

We proposed an explainable reasoning approach for forecasting links on temporal knowledge graphs. The model extracts a query-dependent subgraph from a given temporal KG and performs an attention propagation process to reason on it. Extensive experiments on four benchmark datasets demonstrate the effectiveness of our method. We conducted a survey about the evidence included in the extracted subgraph. The results indicate that the evidence is informative for humans.

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

APPENDIX

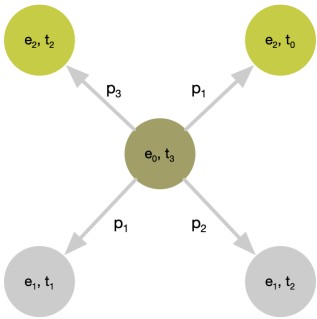

Figure 4: The inference graph for the query $(e_0, p_1, ?, t_3)$. The entity at an arrow's tail, the predicate on the arrow, the entity and the timestamp at the arrow's head build a true quadruple. Specifically, the true quadruples in this graph are as follows: $\{(e_0, p_1, e_1, t_1), (e_0, p_2, e_1, t_2), (e_0, p_3, e_2, t_2), (e_0, p_1, e_2, t_0)\}$. Note that $t_3$ is posterior to $t_0, t_1, t_2$.

---

**Algorithm 1** Reverse Representation Update at the $L^{th}$ Inference Step

**Input:** Inference graph $\mathcal{G}_{inf}$, nodes in the inference graph $\mathcal{V}$, nodes that have been added into $\mathcal{G}_{inf}$ at the $l^{th}$ inference step $\mathcal{V}^l$, sampled prior neighbors $\hat{\mathcal{N}}_v$, hidden representation at the $(L\text{-}1)^{th}$ step $\mathbf{h}_v^{L-1}$, entity embeddings $\mathbf{e}_i$, weight matrices $\mathbf{W}_{sub}^L$, $\mathbf{W}_{obj}^L$, and $\mathbf{W}_h^L$, query $q = (e_q, p_q, ?, t_q)$, update ratio $\gamma$.

**Output:** Hidden representations $\mathbf{h}_v^L$ at the $L^{th}$ inference step, $\forall v \in \mathcal{V}$.

1: **for** $l = L - 1, ..., 0$ **do**
2:     **for** $v \in \mathcal{V}^l$ **do**
3:         **for** $u \in \hat{\mathcal{N}}_v$ **do**
4:             $e_{vu}^L(q, p_k) = \mathbf{W}_{sub}^L(\mathbf{h}_v^{L-1}||\mathbf{p}_k||\mathbf{h}_{e_q}^{L-1}||\mathbf{p}_q)\mathbf{W}_{obj}^L(\mathbf{h}_u^{L-1}||\mathbf{p}_k||\mathbf{h}_{e_q}^{L-1}||\mathbf{p}_q)$,
5:             $\alpha_{vu}^L(q, p_k) = \frac{\exp(e_{vu}^L(q, p_k))}{\sum_{w \in \hat{\mathcal{N}}_v} \sum_{p_z \in \mathcal{P}_{vw}} e_{vw}^L(q, p_z)}$
6:         **end for**
7:         $\widetilde{\mathbf{h}}_v^L(q) = \sum_{u \in \hat{\mathcal{N}}_v} \sum_{o_z \in \mathcal{P}_{vu}} \alpha_{vu}^L(q, p_k)\mathbf{h}_u^{L-1}(q)$,
8:         $\mathbf{h}_v^L(q) = \sigma(\mathbf{W}_h^L(\gamma\mathbf{h}_v^{L-1}(q) + (1 - \gamma)\widetilde{\mathbf{h}}_v^L(q) + \mathbf{b}_h^L))$
9:     **end for**
10: **end for**
11: **Return** $\mathbf{h}_v^L, \forall v \in \mathcal{V}$.

---

| Parameter | Symbol |
|---|---|
| static entity embeddings | $\bar{\mathbf{e}}_i$ |
| frequencies and phase shift of time encoding | $\mathbf{w}, \phi$ |
| predicate embeddings | $\mathbf{p}_k$ |
| weight matrices of TRGA | $\mathbf{W}_{sub}^l, \mathbf{W}_{obj}^l, \mathbf{W}_h^l$ |
| bias vector of TRGA | $\mathbf{b}_h^l$ |
| weight matrix and bias of node embeddings | $\mathbf{W}_v, \mathbf{b}_v$ |

Table 2: Model parameters.

| Sampling Strategies | MRR | Hits@1 | Hits@3 | Hits@10 |
|---|---|---|---|---|
| Uniform | 36.26 | 27.66 | 41.39 | 53.96 |
| Time-aware exponentially weighted | 41.56 | 32.49 | 47.27 | 59.63 |
| Time-aware linearly weighted | 38.21 | 29.25 | 43.77 | 56.07 |
| Last-N-edges | 39.84 | 31.31 | 45.04 | 57.40 |

Table 3: Comparison between model variants with different sampling strategies on ICEWS14 : raw MRR (%) and Hits@1/3/10 (%). In this ablation study, we filter out test triples that contain unseen entities.

## A  RELATED WORK

### A.1  KNOWLEDGE GRAPH MODELS

Representation learning is an expressive and popular paradigm underlying many KG models. The key idea is to embed entities and relations into a low-dimensional vector space. The embedding-based approaches for knowledge graphs can generally be categorized into bilinear models (Nickel et al., 2011; Balažević et al., 2019), translational models (Bordes et al., 2013; Sun et al., 2019), and deep-learning models (Dettmers et al., 2017; Schlichtkrull et al., 2018). Besides, several studies (Hao et al., 2019; Lv et al., 2018; Ma et al., 2017) explore the ontology of entity types and relation types and utilize type-based semantic similarity to produce better knowledge embeddings. However, the above methods lack the ability to use rich temporal dynamics available on temporal knowledge graphs. To this end, several studies have been conducted for link prediction on temporal knowledge graphs (Leblay & Chekol, 2018; García-Durán et al., 2018; Ma et al., 2018b; Dasgupta et al., 2018; Trivedi et al., 2017; Jin et al., 2019; Goel et al., 2019; Lacroix et al., 2020). Ma et al. (2018b) developed extensions of static knowledge graph models by adding timestamp embeddings to their score functions. Besides, García-Durán et al. (2018) suggested a straight forward extension of some existing static knowledge graph models that utilize a recurrent neural network (RNN) to encode predicates with temporal tokens derived from given timestamps. Also, HyTE (Dasgupta et al., 2018) embeds time information in the entity-relation space by arranging a temporal hyperplane to each timestamp. However, these models cannot generalize to unseen timestamps because they only learn embeddings for observed timestamps. Additionally, the methods are largely black-box, lacking the ability to interpret their predictions while our main focus is to employ an integrated transparency mechanism for achieving human-understandable results.

### A.2  EXPLAINABLE REASONING ON KNOWLEDGE GRAPHS

Recently, several explainable reasoning methods for knowledge graphs have been proposed (Das et al., 2017; Xu et al., 2019; Hildebrandt et al., 2020) . Das et al. (2017) proposed a reinforcement learning-based path searching approach to display the query subject and predicate to the agents and let them perform a policy guided walk to the correct object entity. The reasoning paths produced by the agents can explain the prediction results to some extent. Also, Hildebrandt et al. (2020) framed the link prediction task as a debate game between two reinforcement learning agents that extract evidence from knowledge graphs and allow users to understand the decision made by the agents. Besides, and more related to our work, Xu et al. (2019) models a sequential reasoning process by dynamically constructing an input-dependent subgraph. The difference here is that these explainable methods can only deal with static KGs, while our model is designed for forecasting on temporal KGs.

## B  WORKFLOW

We show the workflow of the subgraph reasoning process in Figure 5. The model conducts the reasoning process on a dynamically expanding inference graph $\mathcal{G}_{inf}$ extracted from the temporal KG. This inference graph gives an interpretable graphical explanation about the final prediction. Given a query $q = (e_q, p_q, ?, t_q)$, we initialize the inference graph with the query entity $e_q$ and define the tuple of $(e_q, t_q)$ as the first node in the inference graph (Figure 5a). The inference graph expands by

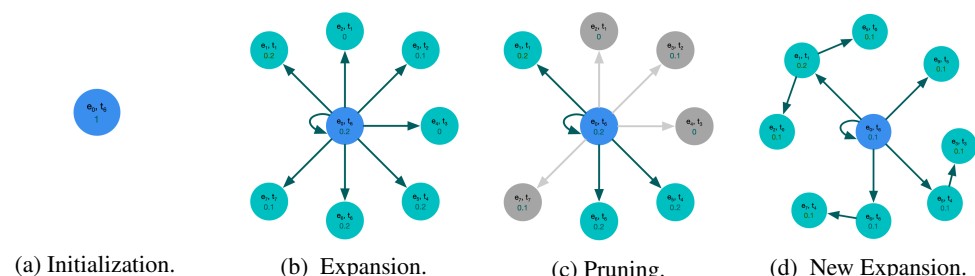

(a) Initialization.   (b) Expansion.   (c) Pruning.   (d) New Expansion.

Figure 5: Inference step by step illustration. Node attention scores are attached to the nodes. Gray nodes are removed by the pruning procedure.

sample neighbors that have been linked with $e_q$ prior to $t_q$, as shown in Figure 5b. The expansion would go rapidly that it covers almost all nodes after a few steps. To prevent the inference graph from exploding, we constrain the number of edge by pruning the edges that are less related to the query (Figure 5c) . Here, we propose a query-dependent temporal relational attention mechanism in Section 4.4 to identify the nodes' importance in the inference graph for query $q$ and aggregate information from nodes' local neighbors. Next, we sample the prior neighbors of the remaining nodes in the inference graph to expand it further, as shown in Figure 5d. As this process iterates, the inference graph incrementally gains more and more information from the temporal KG. After running $L$ inference steps, the model selects the entity with the highest attention score in $\mathcal{G}_{inf}$ as the prediction of the missing query object, where the inference graph itself serves as a graphical explanation.

2

## C DATASET STATISTICS

| Dataset | $N_{train}$ | $N_{valid}$ | $N_{test}$ | $N_{ent}$ | $N_{rel}$ | $N_{timestamp}$ | Time granularity |
|---------|---------|---------|--------|---------|---------|--------------|------------------|
| ICEWS14 | 63685 | 13823 | 13222 | 7128 | 230 | 365 | day |
| ICEWS18 | 373018 | 45995 | 49545 | 23033 | 256 | 304 | day |
| ICEWS0515 | 322958 | 69224 | 69147 | 10488 | 251 | 4017 | day |
| YAGO | 51205 | 10973 | 10973 | 10038 | 10 | 194 | year |

Table 4: Dataset Statistics

| Dataset | $|\mathcal{E}_{tr}|$ | $|\mathcal{E}_{tr}|/|\mathcal{E}|$ | $|\mathcal{E}_{tr+val}|$ | $|\mathcal{E}|$ |
|---------|-------|-------------|--------------|------|
| ICEWS14 | 6180 | 86.7 | 6710 | 7128 |
| ICEWS18 | 21085 | 91.5 | 21995 | 23033 |
| ICEWS0515 | 8853 | 84.4 | 9792 | 10488 |
| YAGO | 7904 | 78.7 | 9008 | 10038 |

Table 5: Unseen entities (new emerging entities) in the validation set and test set. $|\mathcal{E}_{tr}|$ denotes the number of entities in the training set, $|\mathcal{E}_{tr+val}|$ represents the number of entities in the training set and validation set, $|\mathcal{E}|$ denotes the number of entities in the whole dataset.

We provide the statistics of datasets in Table 4. Since we split each dataset into subsets by timestamps, ensuring (timestamps of training set) < (timestamps of validation set) < (timestamps of test set), a considerable amount of entities in test sets is unseen. We report the number of entities in each subset in Table 5.

## D   EVALUATION PROTOCOL

For each quadruple $q = (e_s, p, e_o, t)$ in the test set $\mathcal{G}_{test}$, we create two queries: $(e_s, p, ?, t)$ and $(e_o, p^{-1}, ?, t)$, where $p^{-1}$ denotes the reciprocal relation of $p$. For each query, the model ranks all entities $\mathcal{E}_q^{inf}$ in the final inference graph according to their attention scores. If the ground truth entity does not appear in the final subgraph, we set its rank as $|\mathcal{E}|$ (the number of entities in the dataset). Let $\psi_{e_s}$ and $\psi_{e_o}$ represent the rank for $e_s$ and $e_o$ of the two queries respectively. We evaluate our model using standard metrics across the link prediction literature: *mean reciprocal rank (MRR)*: $\frac{1}{2 \cdot |\mathcal{G}_{test}|} \sum_{q \in \mathcal{G}_{test}} (\frac{1}{\psi_{e_s}} + \frac{1}{\psi_{e_o}})$ and *Hits@k*($k \in \{1, 3, 10\}$): the percentage of times that the true entity candidate appears in the top $k$ of the ranked candidates.

In this paper, we consider two different filtering settings. The first one is following the ranking technique described in Bordes et al. (2013), where we remove from the list of corrupted **triples** all the **triples** that appear either in the training, validation, or test set. We name it *static filtering*. Trivedi et al. (2017), Jin et al. (2019), and Zhu et al. (2020) use this filtering setting for reporting their results on temporal KG forecasting. However, this filtering setting is not appropriate for evaluating the link prediction on temporal KGs. For example, there is a test quadruple (Barack Obama, visit, India, 2015-01-25), and we perform the object prediction (Barack Obama, visit, ?, 2015-01-25). We have observed the quadruple (Barack Obama, visit, Germany, 2013-01-18) in the training set. According to the *static filtering*, (Barack Obama, visit, Germany) will be considered as a genuine triple at the timestamp **2015-01-25** and will be filtered out because the triple (Barack Obama, visit, Germany) appears in the training set in the quadruple (Barack Obama, visit, Germany, 2015-01-18). However, the triple (Barack Obama, visit, Germany) is only temporally valid on 2013-01-18 but not on 2015-01-25. Therefore, we apply another filtering scheme, which is more appropriate for the link forecasting task on temporal KGs. We name it *time-aware filtering*. In this case, we only filter out the triples that are genuine at the timestamp of the query. In other words, if the triple (Barack Obama, visit, Germany) does not appear at the query time of 2015-01-25, the quadruple (Barack Obama, visit, Germany, 2015-01-25) is considered as corrupted and will be filtered out. We report the *time-aware* filtered results of baselines and our model in Table 1.

## E   IMPLEMENTATION

We implement our model and all baselines in PyTorch (Paszke et al., 2019). We tune hyperparameters of our model using a grid search. We set the learning rate to be 0.0002, the batch size to be 128, the inference step $L$ to be 3. Please see the source code[5] for detailed hyperparameter settings. We implement TTransE, TA-TransE/TA-DistMult, and RE-Net based on the code[6] provided in (Jin et al., 2019). We use the released code to implement DE-SimplE[7], TNTComplEx[8], and CyGNet[9]. We use the binary cross-entropy loss to train these baselines and optimize hyperparameters according to MRR on the validation set. Besides, we use the datasets augmented with reciprocal relations to train all baseline models.

## F   REVERSE REPRESENTATION UPDATE MECHANISM FOR SUBGRAPH REASONING

In this section, we explain an additional reason why we have to update node representations along edges selected in previous inference steps. We show our intuition by a simple query in Figure 6 with two inference steps. For simplicity, we do not apply the pruning procedure here. First, we check the equations without updating node representations along previously selected edges. $h_i^l$ denotes the

---

[5]https://github.com/TemporalKGTeam/xERTE
[6]https://github.com/INK-USC/RE-Net
[7]https://github.com/BorealisAI/de-simple
[8]https://github.com/facebookresearch/tkbc
[9]https://github.com/CunchaoZ/CyGNet

hidden representation of node $i$ at the $l^{th}$ inference step.

$$\text{First inference step:} \quad \mathbf{h}_0^1 = f(\mathbf{h}_0^0, \mathbf{h}_1^0, \mathbf{h}_2^0, \mathbf{h}_3^0)$$
$$\mathbf{h}_1^1 = f(\mathbf{h}_1^0)$$
$$\mathbf{h}_2^1 = f(\mathbf{h}_2^0)$$
$$\mathbf{h}_3^1 = f(\mathbf{h}_3^0)$$

$$\text{Second inference step:} \quad \mathbf{h}_0^2 = f(\mathbf{h}_0^1, \mathbf{h}_1^1, \mathbf{h}_2^1, \mathbf{h}_3^1)$$
$$= f(\mathbf{h}_0^1, f(\mathbf{h}_1^0), f(\mathbf{h}_2^0), f(\mathbf{h}_3^0))$$
$$\mathbf{h}_1^2 = f(\mathbf{h}_1^1, \mathbf{h}_4^1, \mathbf{h}_5^1)$$
$$\mathbf{h}_2^2 = f(\mathbf{h}_2^1, \mathbf{h}_7^1, \mathbf{h}_8^1)$$
$$\mathbf{h}_3^2 = f(\mathbf{h}_3^1, \mathbf{h}_6^1)$$

Note that $\mathbf{h}_0^2$ is updated with $\mathbf{h}_1^0, \mathbf{h}_2^0, \mathbf{h}_3^0$ and has nothing to do with $\mathbf{h}_4^1, \mathbf{h}_5^1, \mathbf{h}_6^1, \mathbf{h}_7^1, \mathbf{h}_8^1$, i.e., two-hop neighbors. In comparison, if we update the node representations along previously selected edges, the update in second layer changes to:

$$\text{Second inference step part a:} \quad \mathbf{h}_4^2 = f(f(\mathbf{h}_4^0))$$
$$\mathbf{h}_5^2 = f(f(\mathbf{h}_5^0))$$
$$\mathbf{h}_6^2 = f(f(\mathbf{h}_6^0))$$
$$\mathbf{h}_7^2 = f(f(\mathbf{h}_7^0))$$
$$\mathbf{h}_8^2 = f(f(\mathbf{h}_8^0))$$
$$\text{Second inference step part b:} \quad \mathbf{h}_1^2 = f(\mathbf{h}_1^1, \mathbf{h}_4^2, \mathbf{h}_5^2)$$
$$\mathbf{h}_2^2 = f(\mathbf{h}_2^1, \mathbf{h}_7^2, \mathbf{h}_8^2)$$
$$\mathbf{h}_3^2 = f(\mathbf{h}_3^1, \mathbf{h}_6^2)$$

$$\text{Second inference step part c:} \quad \mathbf{h}_0^2 = f(\mathbf{h}_0^1, \mathbf{h}_1^2, \mathbf{h}_2^2, \mathbf{h}_3^2)$$

Thus, the node 1~3 receive messages from their one-hop prior neighbors, i.e. $\mathbf{h}_1^2 = f(\mathbf{h}_1^1, \mathbf{h}_4^2, \mathbf{h}_5^2)$. Then they pass the information to the query subject (node 0), i.e., $\mathbf{h}_0^2 = f(\mathbf{h}_0^1, \mathbf{h}_1^2, \mathbf{h}_2^2, \mathbf{h}_3^2)$.

## G  SEGMENT OPERATIONS

The degree of entities in temporal KGs, i.e., ICEWS, varies from thousands to a single digit. Thus, the size of inference graphs of each query is also different. To optimize the batch training, we define an array to record all nodes in inference graphs for a batch of queries. Each node is represented by a tuple of (inference graph index, entity index, timestamp, node index). The node index is the unique index to distinguish the same node in different inference graphs.

Note that the inference graphs of two queries may overlap, which means they have the same nodes in their inference graphs. But the query-dependent node representations would be distinct in different inference graphs. To avoid mixing information across different queries, we need to make sure that tensor operations can be applied separately to nodes in different inference graphs. Instead of iterating through each inference graph, we develop a series of segment operations based on matrix multiplication. The segment operations significantly improve time efficiency and reduce the time cost. We report the improvement of time efficiency on ICEWS14 in Table 6. Additionally, we list two examples of segment operations in the following.

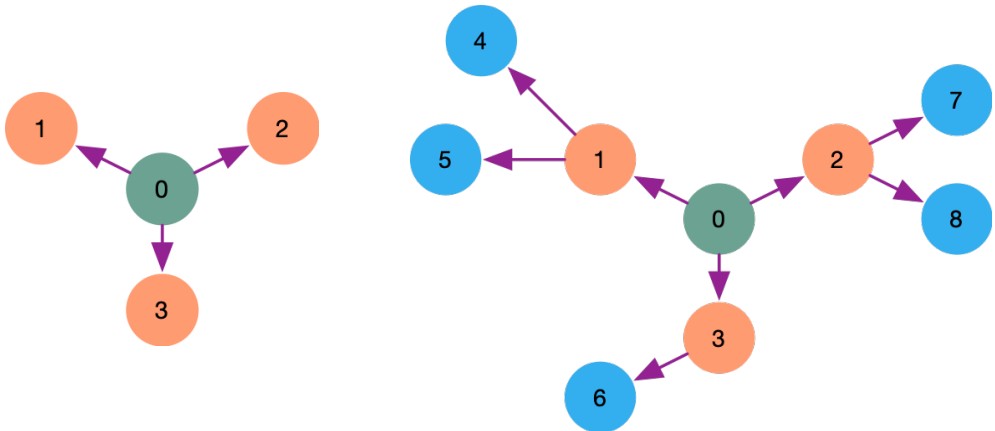

Figure 6: A simple example with two inference steps for illustrating reverse node representation update schema. The graph is initialized with the green node. In the first step (the left figure), orange nodes are sampled; and in the second step (the right figure), blue nodes are sampled. Each directed edge points from a source node to its prior neighbor.

| Computations | Time Cost using Iterator | Time Cost using Segment Operation |
|---|---|---|
| Aggregation of Node Score | 11.75s | 0.004s |
| Aggregation of Entity Score | 3.62s | 0.026s |
| Softmax | 1.56s | 0.017s |
| Node Score Normalization | 0.000738s | 0.000047s |

Table 6: Reduction of time cost for a batch on ICEWS14

**Segment Sum** Given a vector $\mathbf{x} \in \mathbb{R}^d$ and another vector $\mathbf{s} \in \mathbb{R}^d$ that indicates the segment index of each element in $\mathbf{x}$, the segment sum operator returns the summation for each segment. For example, we have $\mathbf{x} = [3, 1, 5]^T$ and $\mathbf{s} = [0, 0, 1]^T$, which means the first two element of $\mathbf{x}$ belong to the $0^{th}$ segment and the last elements belongs to the first segment. The segment sum operator returns $[4, 5]^T$ as the output. It is realized by creating a sparse matrix $\mathbf{Y} \in \mathbb{R}^{n \times d}$, where $n$ denotes the number of segments. We set 1 in positions $\{(\mathbf{s}[i], i), \quad \forall i \in \{0, ..., d\}\}$ of $\mathbf{Y}$ and pad other positions with zeros. Finally, we multiply $\mathbf{Y}$ with $\mathbf{x}$ to get the sum of each segment.

**Segment Softmax** The standard softmax function $\sigma : \mathbb{R}^K \to \mathbb{R}^K$ is defined as:

$$\sigma(\mathbf{z})_i = \frac{\exp(z_i)}{\sum_{j=1}^{K} \exp(z_j)}$$

The segment softmax function has two inputs: $\mathbf{z} \in \mathbb{R}^K$ contains elements to normalize and $\mathbf{s} \in \mathbb{R}^K$ denotes the segment index of each element. It is then defined as:

$$\sigma(\mathbf{z})_i = \frac{\exp(z_i)}{\sum_{j \in \{k | s_k = s_i, \forall k \in \{0, ..., K\}\}} \exp(z_j)}$$

, where $s_i$ denotes the segment that $z_i$ is in.

The segment softmax function can be calculated by the following steps:

1. We apply the exponential function to each element of $\mathbf{z}$ and then apply the segment sum operator to get a denominator vector $\mathbf{d}$. We need broadcast $\mathbf{d}$ such that it aligns with $\mathbf{z}$, which means $\mathbf{d}[i]$ is the summation of segment $\mathbf{s}[i]$.

2. We apply element-wise division between $\mathbf{d}$ and $\mathbf{z}$.

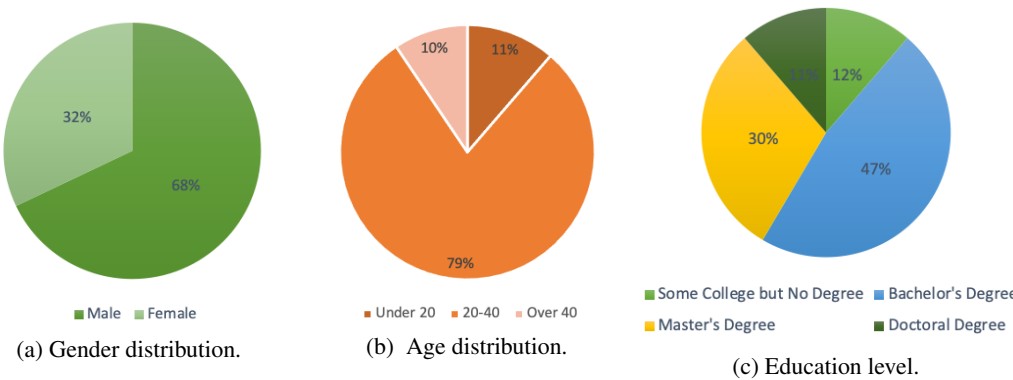

(a) Gender distribution.  (b) Age distribution.  (c) Education level.

Figure 7: Information about the respondent population.

## H SURVEY

In this section, we provide the online survey (see Section 5.3 in the main body) and the evaluation statistics based on 53 respondents. To avoid biasing the respondents, we did not inform them about the type of our project. Further, all questions are permuted at random.

We set up the quiz consisting of 7 rounds. In each round, we sample a query from the test set of ICEWS14/ICEWS0515. Along with the query and the ground-truth object, we present the users with two pieces of evidence extracted from the inference graph with high contribution scores and two pieces of evidence with low contribution scores in randomized order. The respondents are supposed to judge the relevance of the evidence to the query in two levels, namely relevant or less relevant. There are three questions in each round that ask the participants to give the most relevant evidence, the most irrelevant evidence, and rank the four pieces of evidence according to their relevance. The answer to the first question is classified as correct if a participant gives one of the two statements with high contribution scores as the most relevant evidence. Similarly, the answer to the second question is classified as correct if the participant gives one of the two statements with low contribution scores as the most irrelevant evidence. For the relevance ranking task, the answer is right if the participant ranks the two statements with high contribution scores higher than the two statements with low contribution scores.

### H.1 POPULATION

We provide the information about gender, age, and education level of the respondents in Figure 7.

### H.2 AI QUIZ

You will participate in a quiz consisting of eight rounds. Each round is centered around an international event. Along with the event, we also show you four reasons that explain why the given event happened. While some evidence may be informative and explain the occurrence of this event, others may irrelevant to this event. Your task is to find the most relevant evidence and most irrelevant evidence, and then sort all four evidence according to their relevance. Don't worry if you feel that you cannot make an informed decision: Guessing is part of this game!

Additional Remarks: Please don't look for external information (e.g., Google, Wikipedia) or talk to other respondents about the quiz. But you are allowed to use a dictionary if you need vocabulary clarifications.

**Example**

Given an event, please rank the followed evidence according to the relevance to the given event. Especially, please select the most relevant reason, the most irrelevant reason, and rank the relevance from high to low.

*Event: French government made an optimistic comment about China on 2014-11-24.*

A. First, on 2014-11-20, South Africa engaged in diplomatic cooperation with Morocco. Later, on 2014-11-21, a representative of the Morocco government met a representative of the French government.

B. First, on 2014-11-18, the Chinese government engaged in negotiation with the Iranian government. Later, on 2014-11-21, a representative of the French government met a representative of the Chinese government.

C. On 2014-11-23, the French hosted a visit by Abdel Fattah Al-Sisi.

D. A representative of the French government met a representative of the Chinese government on 2014-11-21.

*Correct answer*

Most relevant: D      Most irrelevant: A      Relevance ranking: D B C A

**Tasks**

*1. Event: On 2014-12-17, the UN Security Council accused South Sudan.*

A. South Africa engaged in diplomatic cooperation with South Sudan on 2014-12-11.

B. First, on 2014-11-17, Uhuru Muigai Kenyatta accused UN Security Council. Later, on 2014-11-26, the UN Security Council provided military protection to South Sudan.

C. On 2014-12-16, UN Security Council threatened South Sudan with sanctions.

D. South Sudan hosted the visit of John Kerry on 2014-12-16.

Most relevant:      Most irrelevant:      Relevance ranking:

*2. Event: Indonesia police arrested and retained an Indonesia citizen at 2014-12-28.*

A. The Indonesia police claimed that an attorney denounced the citizen on 2014-12-10.

B. Zaini Abdullah endorsed the Indonesia citizen on 2014-12-25.

C. The Indonesia police made an optimistic comment on the citizen on 2014-12-14.

D. The Indonesia police investigated the citizen on 2014-12-08.

Most relevant:      Most irrelevant:      Relevance ranking:

*3. Event: A citizen from Greece protested violently against the police of Greece on 2014-11-17.*

A. The Greek head of government accused the political party "Coalition of the Radical Left" on 2014-05-25.

B. Greek police refused to surrender to the Greek head of government on 2014-10-15.

C. Greek citizens gathered support on behalf of John Kerry on 2014-11-17.

D. Greek police arrested and detained another Greek police officer on 2014-11-04.

Most relevant:      Most irrelevant:      Relevance ranking:

*4. Event: Raúl Castro signed a formal agreement with Barack Obama on 2014-12-17.*

A. First, on 2009-01-28, Dmitry A. Medvedev made statements to Barack Obama. Later, on 2009-01-30, Raúl Castro negotiated with Dmitry A. Medvedev.

B. Raúl Castro visited Angola on 2009-07-22.

C. Raúl Castro hosted a visit of Evo Morales on 2011-09-19.

D. First, on 2008-11-05, Evo Morales hosted a visit of Barack Obama. Later, on 2011-09-19, Raúl Castro appeal for de-escalation of military engagement to Evo Morales.

Most relevant:      Most irrelevant:      Relevance ranking:

*5. Event: The head of the government of Ukraine considered to make a policy option with Angela Merkel on 2015-07-10.*

A. First, on 2014-07-04, the armed rebel in Ukraine used unconventional violence to the military of Ukraine. Later, on 2014-07-10, the head of government of Ukraine made statements to the armed rebel in Ukraine.

B. The head of the government of Ukraine expressed intent to meet with Angela Merkel on 2014-10-30.

C. First, on 2014-07-04, the armed rebel in Ukraine used unconventional violence to the military of Ukraine. Later, on 2014-07-19, the head of government of Ukraine made statements to the armed rebel in Ukraine.

D. The head of the government of Ukraine consulted with Angela Merkel on 2015-06-06.

Most relevant:     Most irrelevant:     Relevance ranking:

*6. Event: On 2014-08-09, Ukraine police arrested a member of the Ukraine military.*

A. First, on 2014-07-23, a member of Ukraine parliament consulted the head of the Ukraine government. Later, on 2014-07-24, the head of government made a statement to the Ukraine police.

B. First, on 2014-06-25, the military of Ukraine used violence to an armed rebel that occurred in Ukraine. Later, on 2014-07-10, the armed rebel used violence to the Ukraine police.

C. First, on 2005-02-20, the military of Ukraine made a statement to the head of the government of Ukraine. Later, on 2005-07-18, the head of government of Ukraine appealed for a change in leadership of the Ukraine police.

D. On 2014-07-31, the head of the Ukraine government praised the Ukraine police.

Most relevant:     Most irrelevant:     Relevance ranking:

*7. Event: The Office of Business Affairs of Bahrain negotiated with the Labor and Employment Ministry of Bahrain on 2015-07-16.*

A. First, on 2014-07-27, the undersecretary of Bahrain made statements to the Labor and Employment Ministry of Bahrain. Later, on 2015-01-21, an officer of Business Affairs of Bahrain signed a formal agreement with the undersecretary of Bahrain.

B. On 2012-01-21, the office of Business Affairs of Bahrain expressed intent to provide policy support to the employees in Bahrain.

C. First, on 2006-11-01, the employees in Bahrain made statements with the special Rapporteurs of the United Nation. Later, on 2011-05-11, the office of Business Affairs of Bahrain reduced relations with the employees in Bahrain.

D. A representative of the Labor and Employment Ministry of Bahrain consulted with a representative of the Office of Business Affairs of Bahrain on 2014-01-31.

Most relevant:     Most irrelevant:     Relevance ranking:

H.3   GROUND TRUTH ANSWERS

**Question 1**:

Most relevant: B/C      Most irrelevant: A/D

Relevance ranking: BCAD/BCDA/CBAD/CBDA

**Question 2**:

Most relevant: A/D      Most irrelevant: B/C

Relevance ranking: ADBC/ADCB/DABC/DACB

**Question 3**:

Most relevant: B/D      Most irrelevant: A/C

Relevance ranking: BDAC/BDCA/DBAC/DBCA

**Question 4**:

Most relevant: A/D      Most irrelevant: B/C

Relevance ranking: ADBC/ADCB/DABC/DACB

**Question 5**:

Most relevant: B/D      Most irrelevant: A/C

Relevance ranking: BDAC/BDCA/DBAC/DBCA

**Question 6**:

Most relevant: B/C.      Most irrelevant: A/D

Relevance ranking: BCAD/BCDA/CBAD/CBDA

**Question 7**:

Most relevant: A/D      Most irrelevant: B/C

Relevance ranking: ADBC/ADCB/DABC/DACB

## H.4   EVALUATION

The evaluation results of 53 respondents are shown in Figure 8.

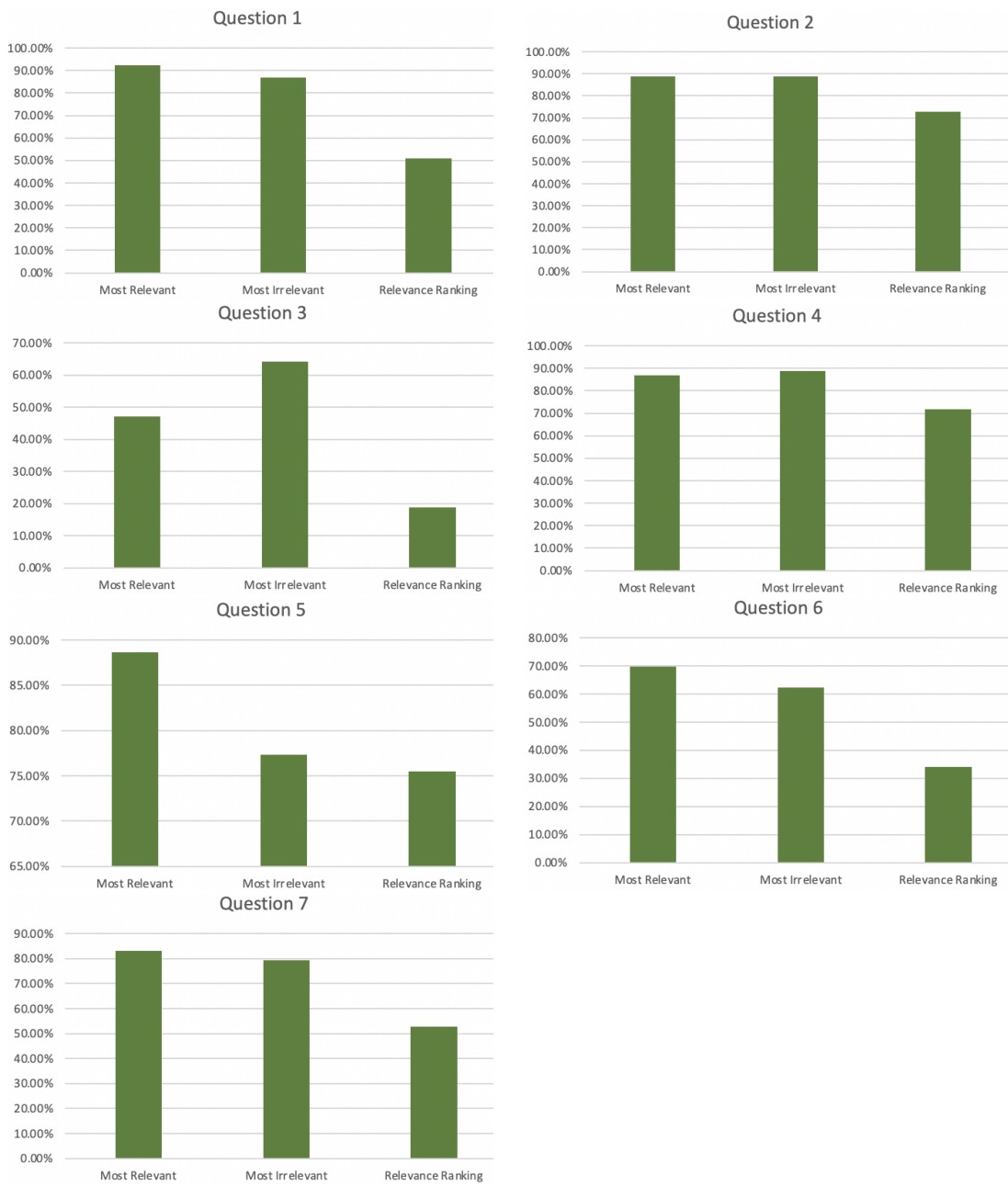

Figure 8: The accuracy of the survey questions.

## I  ADDITIONAL ANALYSIS OF TIME-AWARE ENTITY REPRESENTATIONS

We use a generic time encoding (Xu et al., 2020) defined as $\mathbf{\Phi}(t) = \sqrt{\frac{1}{d}}[\cos(\omega_1 t + \phi_1), \ldots, \cos(\omega_d t + \phi_d)]$ to generate the time-variant part of entity representations (please see Section 4.2 for more details). Time-aware representations have considerable influence on the temporal attention mechanism. To make our point, we conduct a case study and extract the edges' attention scores from the final inference graph. Specifically, we study the attention scores of the interactions between *military* and *student* at different timestamps in terms of the query (*student*, *criticize*, ?, Nov. 17, 2014). We list the results of the model with time encoding in Table 7 and the results of the model without time encoding in Table 8.

As shown in Table 7, by means of the time-encoding, quadruples that even have the same subject, predicate, and object have different attention scores. Specifically, quadruples that occurred recently tend to have higher attention scores. This makes our model more interpretable and effective. For example, given three quadruples {(country A, accuse, country B, $t_1$), (country A, express intent to negotiate with, country B, $t_2$), (country A, cooperate with, country B, $t_3$)}, country A probably has a good relationship with B at $t$ if ($t_1 < t_2 < t_3 < t$) holds. However, there would be a strained relationship between A and B at $t$ if ($t > t_1 > t_2 > t_3$) holds. Thus, we can see that the time information is crucial to the reasoning, and attention values should be time-dependent. In comparison, Table 8 shows that the triple (military, use conventional military force, student) has randomly different attention scores at different timestamps, which is less interpretable.

| Subject | Object | Predicate | Timestamp | Attention Score |
|---|---|---|---|---|
| Military | Student | Use conventional military force | Jan. 17, 2014 | 0.0123 |
| Military | Student | Use conventional military force | May 22, 2014 | 0.0186 |
| Military | Student | Use conventional military force | Aug. 18, 2014 | 0.0235 |
| Military | Student | Use unconventional violence (reciprocal relation) | Aug. 25, 2014 | 0.0348 |

Table 7: Attention scores of the interactions between *military* and *student* at different timestamps (with time encoding).

| Subject | Object | Predicate | Timestamp | Attention Score |
|---|---|---|---|---|
| Military | Student | Use conventional military force | Jan. 17, 2014 | 0.0152 |
| Military | Student | Use conventional military force | May 22, 2014 | 0.0122 |
| Military | Student | Use conventional military force | Aug. 18, 2014 | 0.0159 |
| Military | Student | Use unconventional violence (reciprocal relation) | Aug. 25, 2014 | 0.0021 |

Table 8: Attention scores of the interactions between *military* and *student* at different timestamps (without time encoding).

