# OpenReview forum: "Explainable Subgraph Reasoning for Forecasting on Temporal Knowledge Graphs"
_ICLR.cc/2021/Conference — ICLR 2021 Poster_

### Official Review · AnonReviewer1 · 2020-10-21
**Review of the paper**

**Rating:** 6
**Confidence:** 3

**Review:**


Authors have presented a method to forecast future links on temporal knowledge graphs (KGs). They use attention mechanisms to extract a query-dependent subgraph. According to the authors, this extracted subgraph provides a graphical explanation of the prediction. Authors have performed an ablation study to denote the effect of different components (e.g., updating the representation of nodes, time encoding, sampling strategy) in their method. They have tested the performance of their approach on 3 datasets and have shown that their approach outperforms other baselines in terms of Hits and MRR.

Although the method proposed in this work outperforms other baselines, the novelty and contribution of this work is still not completely clear to me. Major contributions are listed in the last paragraph; however,  I believe some of them are derived with straightforward changes to previous works. As an example, authors have mentioned they have provided the "first" explainable model based on attention mechanisms for temporal KGs.

As far as I have understood, by "explainability" authors mean providing the graphical explanation or in other words, providing relevant arguments in the extracted subgraph. The most relevant paper to this line of work is by Xu et al. 2019 which is also cited by authors. Authors explicitly mention that the work by Xu et al. deals with static KGs whereas their method works with temporal KGs as well. To me, the base of both approaches are the same since Xu et al. 2019 also models a reasoning process by constructing a sub-graph and I believe this issue is limiting the novelty of this work. It would be great if authors can show an example in which the work by Xu et al. does not provide a proper explainability while their approach does.

I also recommend that authors define the metrics they have used (i.e., Hits and MRR) either in footnotes or the supplementary material.

The writing of the paper could be improved. Examples are but not limited to:
- Page 3, 2nd paragraph, "with $v_q$ to$
- Page 5, 1st paragraph, "we need pass messages"
- Page 5, 1st paragraph, "$(l − m)$-aeay"
- Page 5, 1st paragraph,  "at l-th inference step"
- Page 6, 1st paragraph, "an unique"

---

> ### Author Response · Authors · 2020-11-25
> **Response to reviewer1, part A**
>
> Thank you very much for your feedback! We appreciate your comments about the explainability and the writing.
>
> **Q1:** It would be great if authors can show an example in which the work by Xu et al. does not provide a proper explainability while their approach does.
>
> **A1:** Thank you very much for your thoughtful comments! The key difference is that our model (xERTE) performs causality-preserved reasoning in the temporal knowledge domain. Specifically, our model is designed for the temporal KG *forecasting* task that aims to predict unknown links at unseen *future* timestamps by utilizing *past* information. Thus, we pose temporal constraints on extracting subgraphs and passing messages during the training and testing to preserve the causal nature of the temporal data, which can be considered as one of our major contributions.  We provide Figure 12 in Appendix K to illustrate the reasoning process, where the vertical position of a node corresponds to the timestamp of its incoming edge. We name a quadruple in a temporal KG an *event*. As shown in Figure 12, giving a query (*citizen, protest, ? May 27*),  xERTE starts from the query subject *citizen* and finds three past events that involved *citizen*, which are $\\{$ (*citizen*, *criticize*, *government*, May 22),  (*citizen*, *denounce*, *assemblies*, May 17), (*citizen*, *make a request*, *court judge*, May 19)$\\}$. Next, for each event, the model further explores which *past* events may trigger it. Taking (*citizen*, *criticize*, *government*, May 22) as an example, the model finds a prior event (*government*, *make statement*, *labor union*, May 15) that may trigger it, so on and so forth.  Finally, we get three chains of reasoning, and the predicted object is *government*. If we start from a leaf node and go to the root node, we can see that each chain strictly follows the chronological order and shows a clear reasoning process. Taking the left chain as an example, (*labor union*, *accuse*, *government*, May 08) may trigger (*government*, *make statement*, *labor union*, May 15) that may lead to (*citizen*, *criticize*, *government*, May 22), and thus, our model predicts (*citizen*, *protest*, *government*, May 27). In comparison,  DPMPN (Xu et al.) is unaware of temporal constraints and cannot provide chronological reasoning chains. Specifically, when extracting a subgraph, after reaching a node $v_i$ via an edge $e_m$, DPMPN simply samples a neighbor of $v_i$ instead of tracing back to find out which past events may trigger $e_m$. Besides, since DPMPN doesn't take time information into account, it may mistakenly utilize future information for reasoning past interactions, which violates the causal nature of the temporal data.
>
> Moreover, as shown in Figure 12 in Appendix K, xERTE bases its prediction on recent events rather than the events that took place long ago, which is aligned with human reasoning behavior. Intuitively, events that took place long ago would have less impact on future events. To follow this intuition, we assign time-aware representations to each entity, which let recent events make a prominent contribution to the prediction. To make our point, we conduct a case study and extract the edges' attention scores from the inference graph. Specifically, we focus on the interactions between *military* and *student* at different timestamps in terms of the query (*student*, *criticize*, ?, Nov. 17, 2014). We list the results of the model with time-aware entity embeddings in Table 9 in Appendix J and the results of the model with stationary embeddings in Table 10. As shown in Table 9, by means of the time-aware embeddings, quadruples, which even have the same subject, predicate, and object, have different attention scores. Specifically, quadruples that occurred recently tend to have higher attention scores. This makes our model more interpretable. For example, given three quadruples $\\{($*country A*, *accuse*, *country B*, $t_1)$, $($*country A*, *express intent to negotiate with*, *country B*, $t_2)$,  $($*country A*, *cooperate with*, *country B*, $t_3)\\}$, country A would have a good relationship with B at $t$ if $(t_1 < t_2 < t_3 < t)$ holds. But there would be a strained relationship between A and B at $t$ if $(t > t_1 > t_2 > t_3)$ holds. Thus, we can see that the time information is crucial to the reasoning, and attention values should be time-dependent. In comparison, Table 10 (in Appendix J) shows the results of the model with stationary entity embeddings. We see that the triple (military, use conventional military force, student) has randomly different attention scores at different timestamps, which is less interpretable.

---

> ### Author Response · Authors · 2020-11-25
> **Response to reviewer1, part B**
>
> **Q2:** I also recommend that authors define the metrics they have used (i.e., Hits and MRR) either in footnotes or the supplementary material.
>
> **A2:** Thank you for your question. The definitions of the metrics (i.e., Hits and MRR) are in the first paragraph of Appendix D (*Evaluation protocol*).
>
>
> **Q3:** The writing of the paper could be improved. Examples are but not limited to the follows.
>
> **A3:** Thank you very much for pointing them out. We have fixed them and refined the paper.

---

### Official Review · AnonReviewer4 · 2020-10-27
**This paper violates the rule of anonymity**

**Rating:** 1
**Confidence:** 3

**Review:**

In Appendix I.1 it says "We are machine learning scientists from the University of Munich...", which violates the rule of anonymity. Please let me know if I should continue on reviewing this paper. Thanks!

---

### Official Review · AnonReviewer3 · 2020-10-27
**Temporal Knowledge Graph forecasting is an important problem that rises in many applications, and the authors have used an array of attention mechanisms and subgraph sampling techniques to tackle the problem. The clarity of their presentation however needs improvement, and some details further clarifications.**

**Rating:** 6
**Confidence:** 4

**Review:**

This paper addresses the problem of link forecasting in temporal knowledge graphs with special emphasis on generating explainable predictions. The explainability component of their work involves producing a subgraph of the knowledge graph (called the inference graph) that serves as a context of the query being asked, and on which the prediction decision was based.

The inference graph is generated by sampling a subset of prior edges in the L-hop neighborhood of the query node. The authors try different sampling schemes to select a subset of edges. They then use a GNN aggregation scheme to produce a time-away representation of each node. This representation is generated by taking into account the “relevance” of nodes and edges to the query by employing a relational attention mechanism. Each node is then given a plausibility score, and the node with highest plausibility score is selected to be the answer to the query.

The authors run experiments on standard temporal datasets and have appropriate ablation studies. They compare their work to existing methods, and their experiments show that their model performs significantly better than state of the art.
Link forecasting in temporal knowledge graphs is an important problem that arises in many real-life applications, and being able to explain these predictions is equally important. The model proposed by the authors contain the right components for tackling the problem: subgraph sampling based on time, attention mechanism based on the role of a given entity in the query, message passing to learn from neighborhood information for each node, ...etc.

The writing of the paper however is hard to follow; notation is sometimes incomplete or undefined, mathematical formulations are often unnecessarily complex. The pieces were described in a somewhat topsy-turvy order and It was difficult putting together all the pieces of the algorithm. Some further details below:
- Page 3, paragraph 3: what is tKG? There is possibly a missing word in the sentence before-last of the paragraph “with v_q to .”

- In Definition 2, the definition of N_v and \bar{N_v} is hard to follow, even though the concept is simple. Consider saying it in plain words first, then state the definition of N_v without the union operator (by using an “or” in the conditional)

- There is a lot of repetition in the text. E.g. mention of “sampling the neighbors” a few times in the text until you encounter it in Section 4.5. Perhaps discuss sampling first?)

- In Equation 2 (and elsewhere), what is P_{vw}?

- Section 4.3 was hard to follow. The motivation for computing representations in a reverse manner seems to stem only from computational considerations, but it is not clear whether this reverse update mechanism produces the same representation as the standard “forward” update would. The section in the appendix on the topic did not shed more light to this process.

- In the same paragraph, the sentence starting with “Since we only feed…” is confusing:  is the current inference step m or l ?
Section 4.6: It would be less confusing to put the definition of a_e_i^l before that of a_v^l. Also, at which point is this “plausibility score” computed?

- There is a lot happening in Figure 1 but the figure is not referenced from the text. Why are there two identical modules? Perhaps more explanation would be helpful.

Overall, the paper has a lot of promise and the work conducted looks solid. The authors are encouraged to make another attempt at making their work clearer and understandable.

---

> ### Author Response · Authors · 2020-11-24
> **Response to reviewer3, part A**
>
> Thank you very much for your valuable comments! We appreciate the questions you raised regarding the writing of the paper. We have refined the writing and changed the subsections' order according to the workflow. We show an overview of our model in Section 4.1 and describe the modules in Section 4.2 $\sim$ 4.5.
>
> **Q1(i)**: Page 3, paragraph 3: what is tKG?
>
> **A1(i):** tKG is the abbreviation of the temporal knowledge graph. We have added a sentence to clarify it.
>
> **Q1(ii)**: There is possibly a missing word in the sentence before-last of the paragraph “with v_q to .”
>
> **A1(ii):** Thank you very much for pointing it out. We have fixed it.
>
> **Q2:** In Definition 2, the definition of N_v and \bar{N_v} is hard to follow, even though the concept is simple. Consider saying it in plain words first, then state the definition of N_v without the union operator (by using an “or” in the conditional).
>
> **A2:** Thank you for your suggestion！We have refined it.
>
> **Q3:** There is a lot of repetition in the text. E.g. mention of “sampling the neighbors” a few times in the text until you encounter it in Section 4.5. Perhaps discuss sampling first?)
>
> **A3** Thank you for your suggestion! We have changed the structure and discuss *Neighborhood Sampling* first.
>
> **Q4:** In Equation 2 (and elsewhere), what is $\mathcal P_{vw}$?
>
> **A4:** The definition of $\mathcal P_{vw}$ is below Definition 2 (in Section 2). Specifically, we use $\mathcal P_{uv}$ to represent the set of observed predicates connecting nodes $u$ and $v$. To make it clear, we have reminded this definition below Equation 2.
>
> **Q5 (i):** Section 4.3 was hard to follow. The motivation for computing representations in a reverse manner seems to stem only from computational considerations.
>
> **A5 (i):** Thanks for pointing it out, and sorry for the confusion. We have rewritten Section 4.3. The computational efficiency is one advantage of the reverse representation update. But the primary motivation is to ensure that all existing nodes in an inference graph $\mathcal G_{\textit{inf}}$ can receive messages from nodes that are newly added to $\mathcal G_{\textit{inf}}$ at the current inference step. When humans perform a reasoning process, the internal understanding of entities during the inference and reasoning may change as new arguments join the reasoning process. For example, we want to predict the development trend of company A. We knew that A has the largest market portion, which gives us a high expectation about A. However, a new argument shows that conglomerate B enters this market as a strong competitor. Although the new argument is not directly related to A, it indicates that there will be a high competition between A and B, which lowers our expectations about A. To mimic human reasoning behavior, we should ensure that all existing nodes in $\mathcal G_{\textit{inf}}$ can receive messages from a new node when the new node is added to $\mathcal G_{\textit{inf}}$.
>
> In standard graph neural networks, we update the nodes' representations by aggregating messages from their one-hop neighbors. However, since our subgraph reasoning model (xERTE) adds new nodes into the subgraph at each inference step, the standard message-passing mechanism is not appropriate. Specifically, if we see each inference step as one layer of xERTE, the nodes that are multi-hop neighbors of a new node cannot receive its message. Thus, we propose the novel reverse representation update mechanism for subgraph reasoning. Recall that, to avoid violating temporal constraints, we update the representation of each node by aggregating messages only from its prior neighbors (please see Definition 2 in Section 2 for prior neighbors). And at each inference step, we expand $\mathcal G_{\textit{inf}}$ by adding prior neighbors of existing nodes. For example, assuming that we are at the fourth inference step, for a node that has been added at the second step, we only need to aggregate messages from nodes added at the third and fourth steps. Hence, we can update the representations of nodes in reverse order as they have been added in $\mathcal G_{\textit{inf}}$. Specifically, at the $l^{th}$ inference step, we first update the representations of nodes added at the $(l-1)^{th}$ inference step, then the nodes added at $(l-2)^{th}$, and so forth until $l = 0$, as shown in Algorithm 1 in the appendix.

---

> ### Author Response · Authors · 2020-11-24
> **Response to reviewer3, part B**
>
> **Q5(ii):** It is not clear whether this reverse update mechanism produces the same representation as the standard “forward” update would. The section in the appendix on the topic did not shed more light to this process.
>
> **A5(ii):** The representation produced by the reverse update mechanism is not the same as the standard "forward" update would. In the following, we compare the representation update of a standard graph neural network with ours. Figure 8 in Appendix F shows an inference graph where the green node is the query subject (node 0), the orange nodes (node 1/2/3) are its one-hop prior neighbors sampled at the first inference step, and the blue nodes (node 4 - 8) are two-hop prior neighbors sampled at the second inference step. First, we check the message passing of a standard graph neural network. We denote the representation update function as $f(\cdot)$. Note that, to avoid violating temporal constraints, we update the representation of a node by aggregating messages only from its prior neighbors. At the *first* inference step, the representation of the query subject (node 0) is $\mathbf h^1_0 = f(\mathbf h^0_0, \mathbf h^0_1, \mathbf h^0_2, \mathbf h^0_3)$, where the superscript denotes the inference step, and the subscript denotes the node index.  Since the orange nodes don't have prior neighbors at the first inference step, their representations are updated as $\mathbf h_1^1 = f(\mathbf h^0_1), \mathbf h_1^2 = f(\mathbf h^0_2), \mathbf h_1^3 = f(\mathbf h^0_3)$. At the *second* inference step, the query subject (node 0) has the following representation: $\mathbf {h}^2_0 = f(\mathbf {h}^1_0, \mathbf {h}^1_1, \mathbf{h}^1_2, \mathbf{h}^1_3)= f( f(\mathbf h^0_0, \mathbf h^0_1, \mathbf h^0_2, \mathbf h^0_3), f(\mathbf{h}^0_1), f(\mathbf{h}^0_2), f(\mathbf{h}^0_3))$. We can see the query subject (node 0) doesn't receive any message from the blue nodes (node 4$\sim$8). Next, we show the representation update using the proposed reverse mechanism (Algorithm 1). Taking the *second* inference step as an example, we update the node representations in the reverse order in which the nodes were added in the inference graph. In other words, we first update nodes 4$\sim$8 that were added at the second inference step, then nodes 1$\sim$3 that were added at the first step, and finally the query subject (node 0). Thus, nodes 1$\sim$3 receive messages from nodes 4~8, i.e. $\mathbf{h}^{2}_1 = f(\mathbf{h}^1_1, \mathbf{h}^2_4, \mathbf{h}^2_5)$. Then they can pass the information to the query subject (node 0), i.e., $\mathbf{h}^{2}_0 = f(\mathbf{h}^{1}_0, \mathbf{h}^{2}_1, \mathbf{h}^{2}_2, \mathbf{h}^{2}_3)$.
>
> Besides, empirical results show that the model with the reverse update performs better than that with a standard representation update. Since the reverse representation update ensures that each node can receive messages from all its multi-hop prior neighbors in the inference graph $\mathcal G_{\textit{inf}}$, we expect this mechanism could help nodes mine available information. This update mechanism should be especially important for nodes that only have been involved in a small number of events. Since the historical information of such nodes is quite limited, it is very challenging to forecast their future behavior. In Figures 2(a) and 2(b), we show the metrics of Hits@1 and Hits@10 against the number of entities in the inference graph. It can be observed the model with the reverse update mechanism performs better in general. In particular, this update mechanism significantly improves the performance if the query subject only has a small number of (multi-hop) neighbors in the inference graph, which meets our expectation.

---

> ### Author Response · Authors · 2020-11-24
> **Response to reviewer3, part C**
>
> **Q6(i):** In the same paragraph, the sentence starting with “Since we only feed…” is confusing: is the current inference step m or l ?
>
> **A6(i):** Thank you for pointing it out. The current inference step is $l$. We have rewritten the whole paragraph and deleted the confusing sentences.
>
> **Q6(ii):** Section 4.6: It would be less confusing to put the definition of a_e_i^l before that of a_v^l.
>
> **A6(ii):**  Thank you! We have followed your suggestion.
>
> **Q6(iii):** Also, at which point is this “plausibility score” computed?
>
> **A6(iii):** As in many knowledge graph models, we compute the plausibility score in the loss function and train the model such that the ground truth has a high plausibility score and false answers have low plausibility scores. More concretely, given a query $q  = (s, p, ?, t)$, our model extracts an inference graph from the temporal KG and computes a plausibility score $a_{e_i, q}$ for each entity in the extracted graph.  In the training phase, we use the binary cross-entropy loss  $\mathcal L =  - \frac{1}{|\mathcal Q|}\sum_{q\in\mathcal Q}\frac{1}{|\mathcal E_q^{inf}|}\sum_{e_i \in \mathcal E_q^{inf}}\left(y_{ e_i, q}\log(a_{e_i,q}) + (1- y_{e_i, q}) \log(1-a_{e_i, q})\right)$, where $\mathcal E_q^{inf}$ represents the set of entities in the inference graph regarding $q$, $y_{e_i, q}$ represents a binary label that indicates whether $e_i$ is the answer for $q$, and $\mathcal Q$ denotes the set of training quadruples. In the test phase, we rank entities in an inference graph by their plausibility score and choose the top one as our prediction.
>
> **Q7:** There is a lot happening in Figure 1 but the figure is not referenced from the text. Why are there two identical modules? Perhaps more explanation would be helpful.
>
> **A7:** Thank you very much for your suggestion. We have refined Figure 1 and added detailed explanations. Specifically, we show the information flow between two inference steps in Figure 1. Thus, there are two identical modules where the left module indicates the first inference step and the right module indicates the second inference step. The *Attention* block has been explained in Section 4.3. We explain the *Embedding* block in Sections 4.4 and 4.5.  *Score update & pruning* and *Aggregation* are referenced from Section 4.6.

---

### Official Review · AnonReviewer2 · 2020-10-28
**Time awareness for link prediction in knowledge graphs with an eye on interpretability**

**Rating:** 6
**Confidence:** 4

**Review:**


**[UPDATE, 30 Nov]: Rating raised after reading the authors rebuttal.**

The paper presents a knowledge graph embedding model that learns latent representation of a temporal knowledge graph to predict unseen events.

The problem addressed in the paper is relevant for the community, and the angle proposed by the authors is interesting, i.e. tackling learning from temporal graphs with an eye to interpretability.

The paper is well structured, although some sections lack clarity and at times makes reading difficult (e.g. 4.3).

Related work is comprehensive and gives a good coverage of recent temporal KGE works.

The choices made in the architecture (which is grounded on the so-called “inference graph”) seem reasonable.

It is worth mentioning that is the first KGE paper I see that includes a user evaluation survey. Given the narrative emphasises explainability, this is certainly appreciated.

Experiments use agreed-upon datasets and evaluation protocol.

Nevertheless, there are some shortcomings. I have some doubts and questions for the authors:

1. Why not replacing Table 1 with Table 7 (appendix)? Reporting raw metrics in the main paper is less meaningful that the time-aware filtered settings.
2. Baselines: experiments in table 1 do not include recent SOTA-level models such as DE-Simple [Goel 2020] and TNTComplex [Lacroix 2020] - this despite both works are present in the bibliography. Given that xERTE claims to beat SOTA, I believe it is necessary to add those two models in the experiments to make sure SOTA is beaten, in particular using the time-aware filtered settings (Table 7, appendix).
3. The paper does not include experiments on training time. It will be interesting to empirically validate claims of scalability. Training time is a deal breaker for time-aware models, so reporting figures would really add value to the work.
4. The reverse representation update mechanism in 4.3 lacks clarity. Could you please re-phrase and provide a bit more clarity? I fail to see how what described “mimics human behaviour”.
5. In a note on page 3 you say reciprocal triples are added. How does this impact training time, given that in practice you double the size of the training set?
6. In 4.1, how is the number of expansion sets L chosen? Is that an hyperparameter? In that case, it would have been interesting to see some experiments of predictive power and training time.
7. What is the role of the \gamma hyperparameter in 4.2? Have you provided some experiments to show what happens with ranges of different values?
8. Have you run ablation studies on the attention mechanism? (i.e. turning it on/off).
9. How is the time encoding \Phi defined (4.4)? Could you please clarify?
10. Unfortunately, the human evaluation is dismissed in the appendix. While I appreciate the effort of running a user survey,  the summary proposed in 5.3 lacks clarity and this piece of your work would deserve more prominence.
11. Human evaluation: I have doubts on using only 7 questions in the questionnaire. I wonder how meaningful results are. Besides, the questionnaire does not compare against a baseline, so it is hard to tell if the users are really satisfied with the content of the inference graph.
12. Human evaluation: could you add some details on the population? (i.e. the 53 users)

Minor comments:
- figures 2,3 are too small.
- content of user survey description in I.1 may partially break anonymity.

---

> ### Author Response · Authors · 2020-11-20
> **Response to reviewer2, part A**
>
> Thank you very much for your helpful feedback and valuable comments!  We appreciate the questions you raised regarding experiments and human evaluation. We are working on resolving the issues and incorporating the comments in our updated draft.
>
> **Q1**: Why not replacing Table 1 with Table 7 (appendix)? Reporting raw metrics in the main paper is less meaningful than the time-aware filtered settings.
>
> **A1**: Thank you for pointing it out. We have replaced Table 1 with Table 7.
>
> **Q2**: Baselines: experiments in table 1 do not include recent SOTA-level models such as DE-Simple [Goel 2020] and TNTComplex [Lacroix 2020] - this despite both works are present in the bibliography. Given that xERTE claims to beat SOTA, I believe it is necessary to add those two models in the experiments to make sure SOTA is beaten, in particular using the time-aware filtered settings (Table 7, appendix).
>
> **A2**: Thank you for asking for additional comparisons to help validate our claims. We have updated Table 1 with the filtered results of DE-SipmlE and TNTComplex. We can see that our model beats them on all four datasets. DE-Simple and TNTComplex are designed for the **completion**task on temporal knowledge graphs (tKG). In comparison, our model is proposed for **forecasting**on tKGs. While the completion task aims to predict missing links at observed timestamps, the forecasting task aims to predict unknown links at unseen future timestamps. In other words, for the forecasting task, timestamps of quadruples in the test set are strictly posterior to that in the training set. Taking TNTComplex as an example, it learns distinct embeddings for each timestamp. However, this model cannot well generalize to unseen timestamps because it only learns embeddings of observed timestamps.
>
> **Q3**: The paper does not include experiments on training time. It will be interesting to empirically validate claims of scalability. Training time is a deal breaker for time-aware models, so reporting figures would really add value to the work.
>
> **A3**: Thank you for making an excellent point to strengthen our work! We add a paragraph named *time cost analysis* in Section 5.2 and Figure 4 at the beginning of the appendix to report baselines' training time. Overall, our model's training speed is acceptable while being more interpretable and gaining better performance. The model has a comparable training speed with the strongest baseline RE-Net. Specifically, the training time of our model is 1.98 hours on ICEWS14, while RE-Net needs 1.58 hours and TA-TransE needs 2.03 hours.
>
> We have put a lot of effort into reducing the training time of our model. The degree of entities in temporal KGs, i.e., ICEWS, varies from thousands to a single digit. Thus, the size of inference graphs of each query is totally different. To optimize the batch training, we develop a series of *segment operations* and explain them in Appendix H (Segment Operations). By means of the segment operations, the training time is reduced to one-tenth of what they had been on ICEWS14. We report the improvement of each module in Table 8.

---

> ### Author Response · Authors · 2020-11-20
> **Response to reviewer2, part B**
>
> **Q4**: The reverse representation update mechanism in 4.3 lacks clarity. Could you please re-phrase and provide a bit more clarity? I fail to see how what described “mimics human behavior”.
>
> **A4**: When humans perform a reasoning process, the internal understanding of entities during the inference and reasoning may change as new arguments join the reasoning process. For example, we want to predict the tendency of the stock price of company A. We knew that A has the largest market portion, which gives us a high expectation about its stock price. However, a new argument shows that conglomerate B enters this market as a strong competitor. Although the new argument is not directly related to A, it indicates that there will be a high competition between A and B, which lowers our expectation about A. Following this principle, we proposed a reverse representation update rule for subgraph reasoning, ensuring that all existing entities in an inference graph can receive messages from the new entity if a new entity is added into the inference graph. That's what we described as "mimics human behavior".
>
> We show the details of the proposed representation update rule in Algorithm 1 and illustrate it in Figure 8 in Appendix F. In the following, we compare the representation update of a standard graph neural network with ours. Figure 8 shows an inference graph where the green node is the query subject (node 0), the orange nodes (node 1/2/3) are its one-hop prior neighbors sampled at the first inference step, and the blue nodes (node 4$\sim$8) are two-hop prior neighbors sampled at the second inference step. First, we check the message passing of a standard graph neural network. We denote the representation update function as $f(\cdot)$. Note that, to avoid violating temporal constraints, we update the representation of a node by aggregating messages only from its prior neighbors (please see Definition 2 in Section 2 for prior neighbors). At the *first* inference step, the representation of the query subject (node 0) is $\mathbf h^1_0 = f(\mathbf h^0_0, \mathbf h^0_1, \mathbf h^0_2, \mathbf h^0_3)$, where the superscript denotes the inference step, and the subscript denotes the node index.  Since the orange nodes don't have prior neighbors at the first inference step, their representations are updated as $\mathbf h_1^1 = f(\mathbf h^0_1),  \mathbf h_1^2 = f(\mathbf h^0_2),  \mathbf h_1^3 = f(\mathbf h^0_3)$. At the *second* inference step, the query subject (node 0) has the following representation: $\mathbf {h}^2_0 = f(\mathbf {h}^1_0, \mathbf {h}^1_1, \mathbf{h}^1_2, \mathbf{h}^1_3)= f( f(\mathbf h^0_0, \mathbf h^0_1, \mathbf h^0_2, \mathbf h^0_3), f(\mathbf{h}^0_1), f(\mathbf{h}^0_2), f(\mathbf{h}^0_3))$. We can see the query subject (node 0) doesn't receive any message from the blue nodes (node 4$\sim$8). The naive solution is to run the message passing twice at the second inference step. In other words, we need to iterate the message passing $l$ times at the $l$-th inference step to ensure that every node in the inference graph can receive messages from newly added entities. However,  the linear increase of message passing iterations is computationally expensive and contains many unnecessary operations. Next, we show the representation update using the proposed reverse mechanism (Algorithm 1). Taking the *second* inference step as an example, we update the node representations in the reverse order in which the nodes were added in the inference graph. In other words, we first update nodes 4$\sim$8 that were added at the second inference step, then nodes 1$\sim$3 that were added at the first step, and finally the query subject (node 0). Thus, nodes 1$\sim$3 receive messages from nodes 4~8, i.e. $\mathbf{h}^{2}_1 = f(\mathbf{h}^1_1, \mathbf{h}^2_4, \mathbf{h}^2_5)$. Then they can pass the information to the query subject (node 0), i.e., $\mathbf{h}^{2}_0 = f(\mathbf{h}^{1}_0, \mathbf{h}^{2}_1, \mathbf{h}^{2}_2, \mathbf{h}^{2}_3)$. Thus, we pass messages along each edge only once instead of adding a two-layer GNN.

---

> ### Author Response · Authors · 2020-11-20
> **Response to reviewer2, part C**
>
> **Q5**: In a note on page 3 you say reciprocal triples are added. How does this impact training time, given that in practice you double the size of the training set?
>
> **A5**: Thanks for the great question. The additional reciprocal triples do not lead to any real impact on runtime. If we don't add reciprocal triples, we need to run two predictions for each training sample $(s, p, o, t)$: object prediction $(s, p, ?, t)$ and subject prediction $(? , p, o, t)$. Assuming we have $N$ training samples, then we have $2N$ predictions at each epoch. In our work, following the settings in [1],  we add inverse relations and assume that for any relation $p \in \mathcal P$, there exists a relation $p^{−1} \in \mathcal P$ such that $(s, p, o, t) \in \text{tepmoral} \\, \text{KG}$ if and only if $(o, p^{−1}, s, t) \in \text{temporal} \\, \text{KG}$. In this way, we convert subject prediction $(?, p, o, t)$ to object prediction $(o, p^{-1}, ?, t)$ with the reverse relation $p^{-1}$. Thus, we only run object prediction for all training samples including the reciprocal quadruples. The number of predictions is also $2N$.
>
> [1] Balaževič et al. Multi-relational Poincaré Graph Embeddings. Proceedings of the 33rd Conference on NeurIPS, 2019.
>
> **Q6**: In 4.1, how is the number of expansion sets $L$ chosen? Is that a hyperparameter? In that case, it would have been interesting to see some experiments of predictive power and training time.
>
> **A6**: Yes, $L$ is a hyperparameter. Thank you for your suggestion! There is a trade-off between training time and performance. We set $L \in \\{1,2,3,4\\}$ and run experiments on ICEWS14. As shown in Figures 2e and 2f,  the model achieves the best performance with $L=3$ while the inference time considerably increases as $L$ goes up. If seeking efficient computations, we could set $L=2$ to reduce time cost a lot at the price of sacrificing a little performance.
>
> **Q7**: What is the role of the $\gamma$ hyperparameter in 4.2? Have you provided some experiments to show what happens with ranges of different values?
>
> **A7**: The $\gamma$ hyperparameter was introduced against over-smoothing. According to Equations 4 and 5, if $\gamma$ is zero, the old representation of a node $v$ doesn't play any role by its representation update. In other words, the updated representation of $v$ is fully determined by its neighbors. In comparison, if $\gamma$ is larger than zero, the updated representation of $v$ is also related to its old representation. We empirically found that our model's performance degrades if $\gamma$ becomes zero. We provide Figure 6 in the appendix (page 13) to show the model's performance regarding different values of $\gamma$.
>
> **Q8**: Have you run ablation studies on the attention mechanism? (i.e. turning it on/off).
>
> **A8**: Since we use the normalized attention score as the plausibility score to predict the missing object (see Equations 6 and 2), the model degrades to a random sampler if we turn the attention mechanism off. In other words, we randomly choose an entity from the prior neighbors of the query subject as the predicted object. Thus,  the accuracy of a prediction $(s, p, ? ,t_q)$ can be estimated by  $||\mathcal N_{s(p,t_q)}||/||\mathcal N_{s(t<t_q)}||$, where $\mathcal N_{s(t<t_q)}$ denotes entities that have been linked with the subject $s$ before the timestamp $t_q$ , $\mathcal N_{s(p, t_q)}$ denotes the neighbors of $s$ under the predicate $p$ at $t_q$. We do consider a naive baseline and show the results in Tables 5 and 6 in Appendix C. We can see that our model achieves up to 11 times greater Hits@1on ICEWS18 than the naive baseline.
>
> **Q9**: How is the time encoding $\Phi$ defined (4.4)? Could you please clarify?
>
> **A9**: $\boldsymbol \Phi(t)$ represents a generic time encoding [2], which is defined as $\boldsymbol \Phi (t) = \sqrt{\frac{1}{d}}[\cos(\omega_1t), …., \cos(\omega_dt)]$. $d$ denotes the dimensionality of $\boldsymbol \Phi(t)$.  The idea is to inductively learn vector representations of time and model temporal signals by the interactions between the time encoding and structural features.
>
> [2] Xu et al. Inductive representation learning on temporal graphs. ICLR 2020.

---

> ### Author Response · Authors · 2020-11-20
> **Response to reviewer2, part D**
>
> **Q10**: Unfortunately, human evaluation is dismissed in the appendix. While I appreciate the effort of running a user survey, the summary proposed in 5.3 lacks clarity and this piece of your work would deserve more prominence.
>
> **A10**: Thank you for your suggestion! We refined Section 5.3 and described the survey in detail.
>
> **Q11**: Human evaluation: I have doubts on using only 7 questions in the questionnaire. I wonder how meaningful results are. Besides, the questionnaire does not compare against a baseline, so it is hard to tell if the users are really satisfied with the content of the inference graph.
>
> **A11**: Thank you for your feedback! In fact, each question is related to a specific query and have three sub-questions. Specifically, the third sub-question that asks respondents to rank the arguments according to their relevance is quite challenging. But of course, we are happy to conduct a survey with more questions.
> Additionally, since all existing work for temporal knowledge graph reasoning is in a "black-box" fashion and cannot generate an explanation about their prediction, we don't find an appropriate baseline to compare the questionnaire with it.
>
> **Q12**: Human evaluation: could you add some details on the population? (i.e. the 53 users)
>
> **A12**: Thank you for making a great point to strengthen our survey. We have added some figures about the population in Appendix I.1 (Figure 10).

---

### Official Review · AnonReviewer5 · 2020-11-06
**Interesting paper on subgraph inference over temporal KG. Some clarifications needed.**

**Rating:** 7
**Confidence:** 4

**Review:**

**Summary:** This paper proposes xERTE, a comprehensive set of strategies (i.e. a temporal relational attention mechanism and a human-mimic representation update scheme, temporal neighborhood sampling and pruning, etc.) for link forecasting in temporal knowledge graphs (tKGs). Experiments on real-world tKGs show significant improvements and better explainability on KG forecasting.

**Reason to accept:**
(1) Well motivated problem formation and clear definition. (2) Well illustrated model architecture and model details of major components and technical details, the reverse representation update is particularly interesting; (3) Extensive experiments on multiple tKG datasets are conducted. Insights, ablation studies on model variants, and case studies are also provided.

**Reason to reject:**
Model complexity analysis and inference time. Clarity and some details about experiments and sampling techniques.

**Comments & Questions:**
(1) The approach is technically sound. One concern is about model complexity and inference time. The workflow of xERTE seems complicated with multiple attention calculations and representation updates and it would be better to provide the inference time for query (regarding different steps) to show the scalability and whether it is applicable for large-scale tKG.

(2)  Several questions regarding the experiments need clarification:
(i) Is $l$ (step number) or $L$ explicitly given in the paper or manually set up? What is the number of steps for inference in the test set? Also, what is the difference between the expansion step and the inference step?
(ii) How do you make predictions by static KG embedding models by “compressing temporal knowledge graphs into a static, cumulative graph by ignoring the time information”?
(iii) What does “raw MRR/Hits@k” score mean? Does it refer to the raw/filtered settings? If so, why not use filtered evaluation, which seems more reasonable? (iv) The performance of TcomplEx, which is claimed as one baseline, is not reported.
(v) Why are the performance of TA-TransE and T-TransE even worse than static TransE? It was not the case using similar datasets in [4].
(vi) How are all entity embeddings (l=0) and predicated embeddings initialized? Are entity embeddings from 4.4 Dynamic embeddings?  Are all dimension parameters explicitly listed in the paper?

(3) The uniform and time-sensitive sampling are fairly reasonable. However, many KGs (including YAGO used in this paper) have given ontology with specific entity types, domain and range for the edges, which are helpful for KG inference [1,2,3]. Why choose timestamp-weighted sampling instead of sampling by relations or entity types which also significantly limit the subgraph size and are intended to pick high-related relations and important entities.

(4) Can xERTE apply to an inductive setting, in which new entities are emerging as tKG develops as a commonly observed case? A related concern is how the facts in these datasets are split into train/validation/test?

(5) As one major contribution to explainability, it is suggested to provide analysis about justification on temporal model capability. One possible way to do so is to provide how the representation of one node evolves during the inference graph.

(6) In Section 4.4, what are stationary embeddings and time-variant embeddings? Do we assume both are given in the dataset or trained? What is the contribution here to adopt dynamic embeddings?

**Minor issues:**
(1) Almost all figures are not quite visible and the text in the subfigures needs to be enlarged.
(2) Page 3, Inference Graph, Line 7: Incomplete sentence.

**References: **
[1] Ma, S, et al. "Transt: Type-based multiple embedding representations for knowledge graph completion." ECML, 2017.
[2] Lv, X., et al. "Differentiating Concepts and Instances for Knowledge Graph Embedding." Proceedings of the 2018 Conference on EMNLP. 2018.
[3] Hao, J., et al. "Universal representation learning of knowledge bases by jointly embedding instances and ontological concepts." KDD, 2019.
[4] Garcia-Duran, Alberto, Sebastijan Dumančić, and Mathias Niepert. "Learning Sequence Encoders for Temporal Knowledge Graph Completion." EMNLP, 2018.

---

> ### Author Response · Authors · 2020-11-23
> **Response to reviewer5, part A**
>
> Thank you very much for your helpful feedback and valuable comments! We try to resolve the issues and incorporate the comments in our updated draft.
>
> **Q1**: The approach is technically sound. One concern is about model complexity and inference time. The workflow of xERTE seems complicated with multiple attention calculations and representation updates, and it would be better to provide the inference time for query (regarding different steps) to show the scalability and whether it is applicable for large-scale tKG.
>
> **A1**: Overall, the training time of our model is acceptable while being more interpretable and gaining better performance. The model has a comparable training speed to the strongest baseline RE-Net. Specifically, the training time of our model is 1.98 hours on ICEWS14 (with three inference steps), while RE-Net needs 1.58 hours and TA-TransE needs 2.03 hours. We add a paragraph named *time cost analysis* in Section 5.2 and Figure 4 at the beginning of the appendix to report baselines' training time.
>
> The time cost of xERTE is affected not only by the scale of a dataset but also by the number of inference step $L$. We set $L \in \{1,2,3,4\}$ and run experiments on ICEWS14. As shown in Figures 2(e) and 2(f) in Section 5,  the model achieves the best performance with $L = 3$ while the inference time considerably increases as $L$ goes up. If seeking efficient computations, we could set $L=2$ to reduce time cost a lot at the price of sacrificing a little performance.
>
> We have put a lot of effort into reducing the training time of our model. The degree of entities in temporal KGs, i.e., ICEWS, varies from thousands to a single digit. Thus, the size of inference graphs of each query is also different. To optimize the batch training, we develop a series of *segment operations* and explain them in Appendix H (Segment Operations). By means of the segment operations, the training time was reduced to one-tenth of what they had been on ICEWS14. We report the improvement of each module in Table 8.
>
> **Q2**: Several questions regarding the experiments need clarification:
>
> **Q2(i)**: Is $L$ (step number) explicitly given in the paper or manually set up? What is the number of steps for inference in the test set? Also, what is the difference between the expansion step and the inference step?
>
> **A2(i)**: The step number $L$ is a hyperparameter. We have added Figure 2(e) and 2(f) to compare the predictive power with inference time regarding $L$. In addition, we set the number of inference steps in the test set to be the same as that in the training set ($L = 3$).  Moreover, since the inference graph expands only once at each inference step, there is no difference between the expansion step and the inference step.
>
> **Q2(ii)**: How do you make predictions by static KG embedding models by “compressing temporal knowledge graphs into a static, cumulative graph by ignoring the time information”?
>
> **A2(ii)**: Following [1], we extract unique triples (s, p, o) from quadruples (s, p, o, t), and thus, compress temporal KGs into static KGs. Then we could use static KG embedding models to make predictions on the compressed static KGs.
>
> [1] Jin et al. Recurrent Event Network: Autoregressive Structure Inference over Temporal Knowledge Graphs. EMNLP 2020.
>
> **Q2(iii)**: What does “raw MRR/Hits@k” score mean? Does it refer to the raw/filtered settings? If so, why not use filtered evaluation, which seems more reasonable?
>
> **A2(iii)**: Yes, “raw MRR/Hits@k” score refers to the raw/filtered settings. Thank you for your suggestion! We have replaced raw results with filtered results in Table 1.
>
> **Q2(iv)**: The performance of TcomplEx, which is claimed as one baseline, is not reported.
>
>  **A2(iv)**: Thank you for pointing it out! In accordance with the original work of [2], TNTComplEx is an extension of TCopmlEx and has better performance than TComplEx. Therefore, we have added the results of TNTComplEx in Table 1. We can see that our model beats it on all four datasets.
>
> [2] Lacroix et al. Tensor Decompositions for Temporal Knowledge Base Completion

---

> ### Author Response · Authors · 2020-11-23
> **Response to reviewer5, part B**
>
> **Q2(v)**: Why are the performance of TA-TransE and  T-TransE even worse than static TransE? It was not the case using similar datasets in [4].
>
> **A2(v)**: TA-TransE and T-TransE are proposed for the **completion** task on temporal KGs. In comparison, our work focuses on the **forecasting** task. While the completion task aims to predict missing links at observed timestamps, the forecasting task aims to predict unknown links at unseen future timestamps.  In other words, for the forecasting task, timestamps of quadruples in the test set are strictly posterior to that in the training set.
>
> Taking T-TransE as an example, it learns distinct embeddings of each observed timestamp. However, this model cannot well generalize to unseen timestamps. In other words, the embeddings of timestamps in the test set are initialized with random vectors, and they are not learned during the training. T-TransE uses $-||\mathbf s + \mathbf p + \mathbf t - \mathbf o||$  to compute the plausibility score of a quadruple. When performing predictions in the test set, the unlearned timestamp embeddings $\mathbf t$ would interfere with the predictive performance. Thus, the performance of T-TransE is even worse than static TransE for the forecasting task. In comparison,  [4] reports the performance of T-TransE and TA-TransE on the completion task instead of the forecasting task.
>
> Among all baselines that are initially designed for the forecasting task, RE-Net [1] is the strongest one. And our model considerably beats RE-Net on four benchmark datasets of temporal KGs.
>
> [1] Jin et al. Recurrent Event Network: Autoregressive Structure Inference over Temporal Knowledge Graphs. EMNLP 2020.
>
> [4] Garcia-Duran, Alberto, Sebastijan Dumančić, and Mathias Niepert. "Learning Sequence Encoders for Temporal Knowledge Graph Completion." EMNLP, 2018.
>
> **Q2(vi)**: How are all entity embeddings (l=0) and predicated embeddings initialized? Are entity embeddings from 4.4 Dynamic embeddings? Are all dimension parameters explicitly listed in the paper?
>
> **A2(vi)**: The entity and predicate embeddings are initialized with *Xavier initialization*. And yes, the entity embeddings are from Section Dynamic-Embeddings. We list all dimension parameters in Appendix E *Implementation*.
>
> **Q3**: The uniform and time-sensitive sampling are fairly reasonable. However, many KGs (including YAGO used in this paper) have given ontology with specific entity types, domain and range for the edges, which are helpful for KG inference [1,2,3]. Why choose timestamp-weighted sampling instead of sampling by relations or entity types which also significantly limit the subgraph size and are intended to pick high-related relations and important entities.
>
> **A3**: Thank you for making an excellent point to strengthen our model. We have updated the manuscript and cited these papers [1,2,3]. We agree that exploring ontology-weighted sampling is an important direction for our future work. However, most temporal knowledge graph datasets (including the *political* events dataset ICEWS) don't have explicit type information. For example, many entities in ICEWS are personal names, and there are concepts such as "police", "lawyers", and "head of government". If we know which entities (persons) take the role of "head of government", these entities would be the likely candidates of the query (Obama, meets, ?, 2014). But unfortunately, there is no explicit information in ICEWS that aligns a person and an occupation. Thus, it isn't easy to get a cross-view between concepts and instances without external knowledge. Besides, unlike static KGs, many relations in tKG, i.e., "criticize", "cooperate", might relate to a lot of concepts. For instance, "head of government" could criticize both “police” and “lawyers”. Thus, relation-dependent sampling might not be very effective for tKG inference. Moreover, since quadruples in tKG are only valid at some timestamps, the time information is crucial for tKG inference. For example, given three quadruples $\\{($country A, accuses, country B, $t_1)$, $($country A, expresses intent to negotiate with, country B, $t_2)$,  $($country A, cooperates with, country B, $t_3)\\}$, country A probably has a good relationship with B at $t$ if $(t_1 < t_2 < t_3 < t)$ holds. However, there would be a strained relationship between A and B at $t$ if $(t > t_1 > t_2 > t_3)$ holds. Thus, we choose timestamp-weighted sampling.
>
> Overall, we think ontology-based sampling is very interesting and would help tKG inference. But several difficulties need to be overcome in order to incorporate instance-type information. We will continue to work on it.

---

> ### Author Response · Authors · 2020-11-23
> **Response to reviewer5, part C**
>
> **Q4**: Can xERTE apply to an inductive setting, in which new entities are emerging as tKG develops as a commonly observed case? A related concern is how the facts in these datasets are split into train/validation/test?
>
> **A4**: Yes, xERTE can apply to inductive settings. On the one hand, similar to GraphSage [5],  xERTE can infer representations of unseen nodes based on neighbors' representations. On the other hand, xERTE also considers the temporal factor. As tKG evolves, xERTE can infer the time-aware embeddings of observed entities at a future timestamp using a functional time encoding technique (Section 4.5 *Time-aware entity embeddings*). Additionally, our novel reverse representation update mechanism (see Section 4.4 and Figure 2a/2b) helps deal with new entities that only have rare neighbors.
>
> In our experiments, a considerable amount of entities in test sets are new. Taking YAGO as an example, it contains 10038 entities in total, but we only observed 7904 entities in the training set. This is because we sort the quadruples based on their timestamps and split them into subsets by ensuring (timestamps of the training set) < (timestamps of the validation set) < (timestamps of the test set). We have added Table 4 in Appendix C to report the number of entities in each subset. On YAGO, there are 42.8% of test quadruples that contains unseen entities. For such test quadruples, xERTE achieves an MRR of 0.48 and Hits@1 of 42%, indicating its effectiveness in inductive settings. In comparison, existing temporal KG models cannot handle inductive tasks. For example, the strongest baseline RE-Net only achieves an MRR of 0.03 and Hits@1 of 1% for test quadruples that contain unseen entities.
>
> [5] Hamilton et al. Inductive Representation Learning on Large Graphs. NeurIPS 2017.
>
> **Q5**:  As one major contribution to explainability, it is suggested to provide analysis about justification on temporal model capability. One possible way to do so is to provide how the representation of one node evolves during the inference graph.
>
> **A5**: Thank you very much for your suggestion! Since the node representations have high dimensionality, it's hard to visualize their evolution. Specifically, the time-aware node representation is a concatenation of time-invariant entity embeddings and a functional encoding of time. We think the time-aware representation facilitates the temporal model capability and has considerable influence on the temporal attention mechanism. To make our point, we conduct a case study and extract the edges' contribution scores from the final inference graph. Specifically, we study how the interactions between *military* and *student* affect the embeddings of *military* on timestamp Nov. 17, 2014. We list the results of the model with time encoding in Table 9 in Appendix J and the results of the model with stationary embeddings in Table 10. As shown in Table 9, by means of the time-encoding, quadruples, which even have the same subject, predicate, and object, contribute differently to the representation of the temporal node $($*military*, 17/11/2014$)$. Specifically, quadruples that occurred recently tend to have higher contribution scores. This makes our model more interpretable and effective. As the example mentioned in **A3**, given three quadruples $\\{($country A, accuses, country B, $t_1)$, $($country A, expresses intent to negotiate with, country B, $t_2)$,  $($country A, cooperates with, country B, $t_3)\\}$, country A would have a good relationship with B at $t$ if $(t_1 < t_2 < t_3 < t)$ holds. But there would be a strained relationship between A and B at $t$ if $(t > t_1 > t_2 > t_3)$ holds. Thus, we can see that the time information is crucial to the reasoning. The time-dependent embeddings are responsible for capturing the time information and guide our model to focus on more relevant events. In comparison, Table 10 (in Appendix J) shows that the triple (military, use conventional military force, student) has randomly different contribution scores at different timestamps, indicating the lack of temporal model capability.

---

> ### Author Response · Authors · 2020-11-23
> **Response to reviewer5, part D**
>
> **Q6(i)**: In Section 4.4, what are stationary embeddings and time-variant embeddings? Do we assume both are given in the dataset or trained?
>
> **A6(i)**: The embeddings of an entity $e_i$ at time $t$ consist of stationary embeddings and time-variant embeddings. First, stationary embeddings $\mathbf{\bar e}_i$ of an entity $e_i$ are time-invariant. We initialize the stationary embeddings with random vectors and learn them during the training procedure. Besides, we use $\boldsymbol \Phi(t)$ to generate time-variant embeddings.  $\boldsymbol \Phi (\cdot)$ is a continuous time-encoding function [5] defined as $\boldsymbol \Phi (t) = \sqrt{\frac{1}{d}}[\cos(\omega_1t), …., \cos(\omega_dt)]$, where  $\omega_1, .., \omega_t$ are learnable parameters. Thus, $\boldsymbol \Phi (\cdot)$ can inductively infer time-variant embeddings for both new and observed nodes when the graph evolves.
>
> **Q6(ii):** What is the contribution here to adopt dynamic embeddings?
>
> **A6(ii):** Since the graph structures in tKGs are not stationary as edges evolve over time, the dynamic embeddings can help the model capture static node features and temporal patterns. We've also conducted an ablation study to assess the contribution of the dynamic embeddings. As mentioned in Paragraph *time-aware representation analysis* of Section 5.2, the model's performance significantly degrades if removing the time-variant part from entity embeddings. We show the experimental results in Figure 2(c).
>
> As mentioned in the last question's answer, dynamic embeddings also help the model focus on events that happened closer to the query time and let them have a considerable influence on prediction (please see appendix J for more details).
>
> [5] Xu et al. Inductive representation learning on temporal graphs. ICLR 2020.

---

### Decision · Program_Chairs · 2021-01-07
**Final Decision**

**Decision:**

Accept (Poster)

**Comment:**

The paper has received 4 positive reviews all supporting the acceptance of the paper. The authors have provided a strong rebuttal and have addressed the reviewers' concerns. Please make sure to include all reviewer feedbacks in the camera-ready version.